

# The Solar Zenith Angle Impacts MODIS versus CALIPSO AOD Retrieval Biases, with Implications for Arctic Aerosol Seasonality

Sarah Smith[1], Yutian Wu[1], Rob Levy[2], and Mingfang Ting[1]

[1]Lamont-Doherty Earth Observatory, Columbia University, New York, NY, USA
[2]NASA Goddard Space Flight Center, Greenbelt, MD, USA

Corresponding author: Sarah Smith (sarahs@ldeo.columbia.edu)

**Abstract.** Station observations of surface Arctic aerosol have long shown a pronounced seasonal cycle, with burdens characteristically peaking in the late winter/early spring. Cloud-Aerosol Lidar with Orthogonal Polarization (CALIOP) aerosol optical depth (AOD) products replicate this seasonality, but passive sensor and reanalysis data products do not. We find that the sub- and low-Arctic seasonality of gridded AOD products from six passive sensors diverges from that of CALIOP data products during the months of September-April, even when controlling for sampling biases. Using collocated CALIOP and Moderate Resolution Imaging Spectroradiometer (MODIS) (Aqua) retrievals, we find that for collocations characterized by low-quality MODIS retrievals, the bias between MODIS and CALIOP strongly depends on the solar zenith angle (SZA), with MODIS AODs showing a 132% reduction relative to the instrument-mean over a theoretical 0-90° SZA domain. As the fraction of MODIS retrievals flagged as "low-quality" increases with higher SZAs, retrieval quality mediates the relationship between the SZA and dataset biases in gridded products. The dependency is likely the result of cloud-adjacency effects, and likely also affects midlatitude AOD seasonality. Though additional sources of uncertainty in high latitude retrievals remain, the observed dependency impacts passive sensor data products' representations of (sub-)Arctic aerosol burdens in boreal spring and autumn, which are important for understanding aerosol processes in a highly sensitive yet understudied region. This work also contributes to improved understanding and quantification of the effects of viewing geometry on satellite AOD retrievals, which can help constrain aerosol observations and associated forcings, globally.

## 1 Introduction

Aerosol optical depth (AOD) is a column-integrated measure of aerosol-driven light attenuation, and it is closely related to the "direct" aerosol radiative effect. As AOD can be retrieved from spaceborne observations, AOD is a useful metric to assess aerosol burdens over vast and remote geographical areas. In particular, AOD can be inferred from passive sensors that observe reflected solar radiation in one or more spectral bands. Many of these passive sensors observe broad swaths, allowing them



to sample most of the Earth on near-daily timescales. However, deriving AOD from passive sensors is currently difficult or
impossible under many conditions (e.g. at night, under cloudy skies, or over very high surface albedos of ice and snow). As
such, their picture of global aerosol burden is still incomplete, and AOD information from passive sensors is particularly limited
in the high latitudes.

In contrast to passive sensors, LiDAR instruments provide their own light source, deriving AOD from wavelength-specific
backscatter and an inferred LiDAR ratio. As a result, LiDAR instruments can process retrievals at night. Unlike passive sensors,
LiDAR instruments also provide vertically resolved extinction profiles. During the night, surface reflectance affects only the
lowest altitude retrievals, so spaceborne LiDAR can retrieve over high albedo surfaces. Retrievals through optically thin clouds,
and above optically thick clouds, are also possible (Winker et al., 2009). However, the narrow footprint of the LiDAR makes
coverage much sparser, and small-scale, transient aerosol events are less commonly detected (Smith et al., 2022; Mölders
& Friberg, 2023). Still, the ability to retrieve in multiple lighting conditions and over high albedo surfaces makes LiDAR
instruments particularly appealing in the Arctic, where darkness persists for several months during the winter and ice cover
precludes passive sensor retrievals throughout much of the region in spring and autumn.

The Cloud-Aerosol LiDAR with Orthogonal Polarization (CALIOP) aboard the Cloud-Aerosol LIDAR and Infrared Pathfinder
Satellite Observations (Calipso) satellite came online in June of 2006, and was retired in August of 2023. CALIOP was the first
long-term spaceborne LiDAR specifically designed to assess atmospheric aerosols and clouds (Winker et al., 2009). An Earth-
CARE satellite carrying ATmospheric LiDAR (ATLID) was launched in May 2024, and will continue the work of CALIOP
(Haas et al., 2023). CALIOP processes retrievals by identifying and classifying optically homogeneous layers from a backscat-
ter profile as either clean air, cloud, or aerosol; in the version 4 algorithm, aerosol layers are subsequently classified as one
of seven aerosol subtypes (dust, polluted dust, dusty marine, marine, polluted continental, elevated smoke, clean continental),
with each subtype corresponding to an attendant LiDAR ratio (Winker et al., 2009; Kim et al., 2018). Accurate representation
of AODs depend on selecting the correct LiDAR ratio, and so potential errors in aerosol subtyping are a source of uncertainty
(Kim et al., 2018). Globally, CALIOP AODs are biased low relative to MODIS, a result of CALIOP's categorization of opti-
cally thin layers as clean air, when vertically integrated measures from the passive sensors detect statistically significant aerosol
reflectance (Kim et al., 2017; Toth et al., 2018).

CALIOP AODs are flagged as occurring either during the night or day, with attendant, slight differences in processing
algorithms (Kittaka et al., 2011). Daytime AODs show systematically lower values than nighttime AODs, a result of the lower
signal-to-noise ratio (SNR) from solar irradiance during the day (Kittaka et al., 2011). Such biases make examinations of the
seasonality of high latitude AODs challenging—above 60N daytime AODs are infrequent during winter, while nighttime AODs
are absent in midsummer (Di Pierro et al., 2013). Using the version 3 CALIOP data, Di Pierro et al. (2013) find that a linear
scaling factor can be used to calculate a "nighttime-equivalent" AOD value for monthly mean Arctic AODs, though differences
in the high latitude subtyping schemes between versions 3 and 4 bring into question whether the relationship remains consistent
between versions (Kim et al., 2018). In the L3 version 4 CALIOP data, year-round monthly mean AODs are available up to
72N, allowing for characterization of low (but not high) Arctic seasonality with just the CALIOP day data.



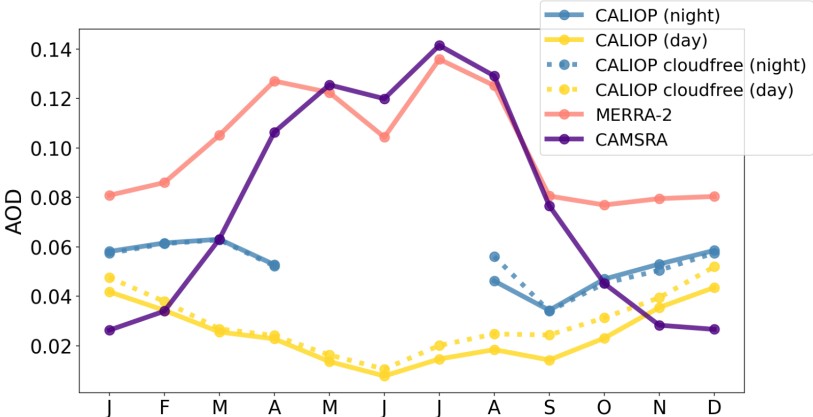

**Figure 1.** Area weighted monthly median AOD from 65-72N for Level 3 CALIOP day (yellow) and night (blue) datasets, for both cloudfree (dashed) and all-sky products (solid), and MERRA-2 (pink) and CAMSRA (purple) reanalysis products from 1/2007-12/2021.

In situ observations of surface-level Arctic aerosol generally show a pronounced seasonal cycle, with burdens characteristically peaking in the late winter/early spring (the so-called "Arctic haze") and reaching a minimum in early to mid-summer
(e.g., Mitchell et al., 1956; Rahn et al., 1977; Shaw et al., 1995). Long-term reductions in European, and more recently East Asian emissions, along with recent increases in boreal wildfire in late summer, have likely dampened the amplitude of the seasonal cycle in recent years (Quinn et al., 2007; Yang et al., 2020; Tao et al., 2020; McCarty et al., 2021; Schmale et al., 2021; Smith et al., 2022). Still, Schmeisser et al. (2018) found that in four of six Arctic stations examined (2012-2014), both absorption and scattering coefficients (550 nm) peaked in the late winter/early spring and exhibited minima in summer, with
the peak an order of magnitude greater than the minima in all such cases. Concentrations of sulfate ($SO_4$) and equivalent black carbon (eBC) at the same stations showed similar seasonality when considered through 2020 (Schmale et al., 2022).

In the Arctic, where daily solar insolation varies substantially from season-to-season, constraining aerosol direct and indirect radiative effects depends on properly constraining aerosol seasonality (Quinn et al., 2008; AMAP, 2015; Zhao et al., 2015). Historically, models have largely failed to represent the seasonality of observations at monitoring stations, though in
the last decade refined parameterizations of black carbon (BC) aging have improved model representations of BC seasonality, specifically (Huang et al., 2010; Eckhardt et al., 2015; Shen et al., 2017). Reanalysis AOD products assimilate observations from passive sensors with modeled transport, offering the promise of widespread coverage even where direct observations are not available. Yet, comparison of low-Arctic (65-72N) AOD from CALIOP night and day data products, with the Modern-Era Retrospective analysis for Research and Applications, Version 2 (MERRA-2) and Copernicus Atmosphere Monitoring Service
(CAMSRA) reanalysis AODs, shows nearly opposite seasonality (Fig. 1). This is true for both the all-sky CALIOP data and the cloud-free data. Given the sensitivity of Arctic temperatures to seasonal variations in radiative forcing, understanding and constraining the seasonality of Arctic AOD is necessary for improved constraints on Arctic warming under various emissions scenarios (Bintanja & Krikken, 2016).





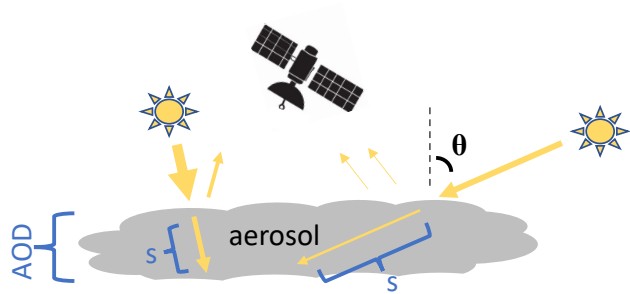

**Figure 2.** A schematic illustration of insolation and reflectance at low (left) and high (right) solar zenith angles ($\theta$), where 's' shows an idealized path through the aerosol. Aerosol Optical Depth (AOD) relates to the light attenuation that would occur along a vertical path through an aerosol, and is inferred from measured reflectance (passive sensors, shown) or backscatter (active sensors).

It is yet unclear the extent to which dataset differences in Arctic AOD seasonality are propagated from assimilated passive
sensor retrievals, or are primarily the result of transport processes in models. Kittaka et al. (2011) found that biases between
CALIOP version 2 and MODIS (Aqua) Collection 5 AODs show notable spatiotemporal variability, with seasonal differences
in the bias particularly evident in the Southern Ocean (the authors attribute this to seasonal differences in cloudiness and wind
speeds, the latter affecting ocean wave glint). Retrieval geometry has also been shown to affect passive sensor retrievals of
cloud optical depth; Maddux et al. (2010) found that MODIS cloud optical depth decreased at more oblique sensor zenith
angles, while Grosvenor & Wood (2014) found that MODIS liquid cloud optical depth increases with the solar zenith angle
(SZA).

At high solar angles, a longer path through the atmosphere results in increased attenuation when aerosol is present (Fig. 2),
as well as more diffuse scattering from neighboring clouds. Passive sensor retrievals could be affected by reduced sensitivity
to higher levels of attenuation (a 1D radiative effect), or biases resulting from scattering within or between clouds and other
reflective surfaces (a 3D radiative effect). Cloud contamination is as well-known source of uncertainty in aerosol remote
sensing, and disentangling cloud artifacts from physical changes to cloud-adjacent aerosols remains an ongoing challenge (e.g.,
Remer et al., 2020). For retrievals over land, the radiative transfer module for the MODIS dark target assumes a plane-parallel
atmosphere, which can cause faulty retrievals at very high SZAs (Vermote & Vermeulen, 1999). Alternatively, SZA-related
differences in the CALIOP SNR could also affect the active sensor's retrieval biases, as could seasonal differences in Arctic
aerosol LiDAR ratios if not accurately selected by the CALIOP subtyping algorithm. Of course, a combination of the above
factors may also explain any observed SZA-related dependencies in the bias between CALIOP and passive sensors.

In this paper we consider three main questions in two sections. In the first section, we examine differences between CALIOP
and passive sensor AOD seasonality in the sub- and low Arctic, and determine the spatial extent of differences between CALIOP
and reanalysis seasonality. Second, after finding that CALIOP and passive sensors show divergent seasonality in the mid-to-
high latitudes, we test the hypothesis that the bias between CALIOP and MODIS retrievals depends on the SZA. Finally, we





examine the correlation between instruments as a function of the SZA, considering the utility of passive sensor data products under different SZA conditions.

## 2 Data

For part 1 of this analysis, we consider Level 3 (L3) data from CALIOP night and day data products as well as from multiple passive sensors. L3 products are aggregated from Level 2 (L2) data, onto a regular spatial and temporal grid; L2 products are processed retrievals derived from the sensors' observations. We also compare CALIOP L3 AODs to reanalysis products from MERRA-2 and CAMSRA.

For part 2 of the analysis, we compare L2 CALIOP daytime AODs from 2010 to collocated AODs from MODIS deep blue dark target combined L2 products.

### 2.1 CALIOP data

CALIOP retrieves backscatter with depolarization at 532 and 1064 nm, with L2 and L3 aerosol data products providing AODs at 532 nm. For comparisons of monthly mean gridded CALIOP versus passive sensor and reanalysis AODs, we acquired the L3 version 4.2 all-sky (CAL_LID_L3_Tropospheric_APro_AllSky-Standard-V4-20/21, 2°x5°x60m) and cloudfree (CAL_LID_L3_Tropospheric_APro_CloudFree-Standard-V4-20/21, 2°x5°x60m) data products. Aerosol layers over opaque clouds are, on average, optically thinner than those extending through the whole column, and as a result the AllSky CALIOP data products (which includes retrievals over optically thick clouds) has a low bias relative to cloudfree CALIOP data products. The 70m footprint of the LiDAR makes for less frequent sampling than passive sensors, with sampling frequencies of ~16 days (Winker et al., 2009). In this analysis we consider the night and day datasets separately.

For collocations with MODIS L2 data, we acquired the CALIOP L2 profile product (CAL_LID_L2_05kmAPro-Standard-V4-20/21). As the L2 CALIOP AODs were collocated with MODIS retrievals, only the daytime products were used. During level 2 processing, backscatter and depolarization ratios from each retrieval are processed using four algorithms: the Selective Iterated BoundarY Locator (SIBYL), which identifies optically homogeneous layers within the profile; the cloud aerosol discrimination algorithm (CAD) which determines whether a layer is aerosol, cloud, or clear sky; the Scene Classification Algorithm (SCA) which classifies aerosol layers into aerosol subtypes; and the Hybrid Extinction Retrieval Algorithms (HERA), which uses the measured backscatter and the LiDAR ratios assigned to each subtype to calculate extinction within an aerosol layer (Winker et al., 2009).

Errors can occur at any level in the processing. However, classification of optically thin aerosol layers as clear sky by the CAD is a well-known source of bias, particularly for daytime CALIOP AODs; CALIOP aerosol extinction coefficients are calculated only for layers in which backscatter is detected over a noise threshold, and which are not otherwise flagged as cloud (Kim et al., 2017; Toth et al., 2018). CALIOP profile bins corresponding to layers in which backscatter falls below the noise threshold are assigned a 'retrieval fill value,' (RFV) and do not contribute to column integrated AODs (Toth et al., 2018). Thus, a profile in which all layers are assigned an RFV will result in an AOD of zero, even if some aerosol is present below detection



thresholds (Kim et al., 2017; Toth et al., 2018). Scattering from sunlight increases background reflectance, making the noise threshold higher for daytime versus nighttime CALIOP retrievals, and resulting in systematically lower daytime AODs.

Using the version 3 CALIOP data, Toth et al. (2018) found that 71% of all daytime profiles, and 45% of cloudfree daytime profiles, consisted of so-called 'all-RFV' profiles, in which an AOD of zero was detected. MODIS Dark Target retrievals collocated with CALIOP all-RFV profiles showed a mean value of .06 (.08 for AERONET collocations), indicating that column-integrated measures detect non-negligible AOD in cases where the noise threshold for CALIOP aerosol layer detection is not surpassed.

Misidentification of aerosol subtype, or incorrect LiDAR ratios, are additional sources of uncertainty (Kim et al., 2018). Globally, CALIOP AODs have been validated against AERONET, with a low bias (mean absolute error of -.051) (Kim et al., 2018).

## 2.2 AVHRR data

The first in a series of Advanced Very High Resolution Radiometers (AVHRR) was launched in 1981, making AVHRR data
products the longest-running satellite AOD datasets. AVHRR retrievals over ocean are processed using the Satellite Ocean Aerosol Retrieval (SOAR) algorithm (Hsu et al., 2017), which have been validated against MODIS and SeaWiFS AOD data products (Sayer et al., 2017). We use the Monthly AVHRR Aerosol Optical Thickness (650 nm) over Global Oceans version 4.0 dataset (AOT_AVHRR_v04r00_monthly-avg), at .1°x.1° resolution.

## 2.3 MISR data

The Multi-angle Imaging SpectroRadiometer (MISR), which flies aboard Terra, measures reflectance from nine distinct viewing angles, allowing for improved constraints on surface reflectance and differentiation between particle types (Diner et al., 1998). We use the 555 nm optical depth from all particle types in the MISR L3 Component Global Aerosol product, version 4 (MIL3MAEN_4), at .5°x.5° resolution.

## 2.4 MODIS data

We use L2 and L3 MODIS (Aqua) AODs, from the MODIS Collection 6.1 (MYD04_L2 and MYD08_M3 1°x1°, respectively), with AODs at 550 nm. MODIS (Aqua) has been in use since 2002, and retrieves reflectance in 36 spectral bands (Levy et al., 2013). A viewing swath width of 2330 km allows for near-daily observations over most of the Earth (Levy et al., 2013). For both the L3 and L2 products, we use the combined Deep Blue Dark Target (DBDT) aerosol product (Sayer et al., 2014). Both constituent and the merged products have been extensively validated, with DBDT AODs in MODIS Collection 6.1 showing a
high bias relative to AERONET observations (mean absolute errors of .067, globally) (Wei et al., 2019). The radiative transfer for the dark target algorithm assumes a pseudospherical atmosphere over ocean, but plane-parallel over land, while the deep blue algorithm uses a pseudospherical atmosphere for all retrievals.



Since Collection 5, MODIS processing algorithms have allowed retrievals of negative AODs (values as low as -.05) (Levy et al., 2007). MODIS optical depth retrievals depend on correct calibration of surface reflectance and aerosol properties, which can be both overestimated and underestimated. Without including negative AOD values, the dataset would selectively exclude the effects of overestimating–but not underestimating–noise in the MODIS dataset, biasing the dataset high (Levy et al., 2007). Thus, while negative optical depths are physically meaningless, low magnitude, negative AODs in the MODIS dataset are included in the L2 retrievals, and are considered functionally similar to an AOD of zero (Levy et al., 2007).

MODIS L2 AODs are accompanied by quality control (QC) flags. The values for QC flags range from 0 ('bad') to 3 ('good'). QC flags of 1 are typically assigned in scenes characterized by relatively few valid pixels for a given scan; pixels contaminated by clouds, ocean glint, snow or ice, inland water, or high coastal sedimentation are likely to be screened (Levy et al., 2024). L3 products utilize different averaging schemes to produce monthly mean gridded products; the L3 combined DBDT product is prefiltered to remove the lowest quality (QC flag = '0') retrievals, and the remaining retrievals are averaged without further regards to retrieval quality.

The frequency of low-quality to high-quality retrievals is not uniform in space and time, with low quality retrievals more common above 60N from the months of October to March (Fig. 3). Similarly, retrievals over the Southern Ocean are more likely to be characterized by low QC flags, likely the result of the glint over the area. We are particularly interested in understanding biases in the L3 AOD products in areas characterized by an increased reliance on low quality retrievals. As such, our analysis in part two is applied to valid L2 retrievals with QC flags ranging from 1-3 (we exclude retrievals with a QC flag of 0 from the analysis, consistent with L3 processing).

## 2.5 POLDER data

The third iteration of the Polarization and Directionality of the Earth's Reflectances (POLDER) instrument, aboard the Polarization & Anisotropy of Reflectances for Atmospheric Sciences coupled with Observations from a Lidar (Parasol) Satellite, flew in the A-train from 2005-2013. POLDER collects reflectance from 12 different viewing angles across eight spectral bands, three of which provide polarized reflectances (Deschamps et al., 1994). The retrievals are processed using the Generalized Retrieval of Atmosphere and Surface Properties (GRASP) "Models" algorithm, version 2.1, with L3 products gridded to 1x1° resolution (GRASP_POLDER_L3) (Dubovik et al., 2011, 2014, 2021). We use POLDER AODs at 565 nm in this analysis. Globally, the PARASOL/GRASP (Models) algorithm provides the greatest agreement with AERONET of the three POLDER/GRASP algorithms assessed in Chen et al. (2020), with R=.920 and .950 and root mean square errors (RMSEs) of .120 and .048 for 565nm over land and ocean, respectively.

## 2.6 SeaWiFS data

We use the L3 Deep Blue AOD Angstrom Exponent product from Sea-viewing Wide Field-of-view Sensor (SeaWiFS), version 004 (SWDB_L3M10, 1x1°) at 550 nm. The instrument was active from 1997-2010, and detects reflectance over eight spectral channels (Sayer et al., 2012a). The Deep Blue algorithm, which is also used in processing MODIS and VIIRS AODs, was applied to the SeaWiFS ocean and land retrievals (Sayer et al., 2012a; Hsu et al., 2013). Validation against Maritime Aerosol





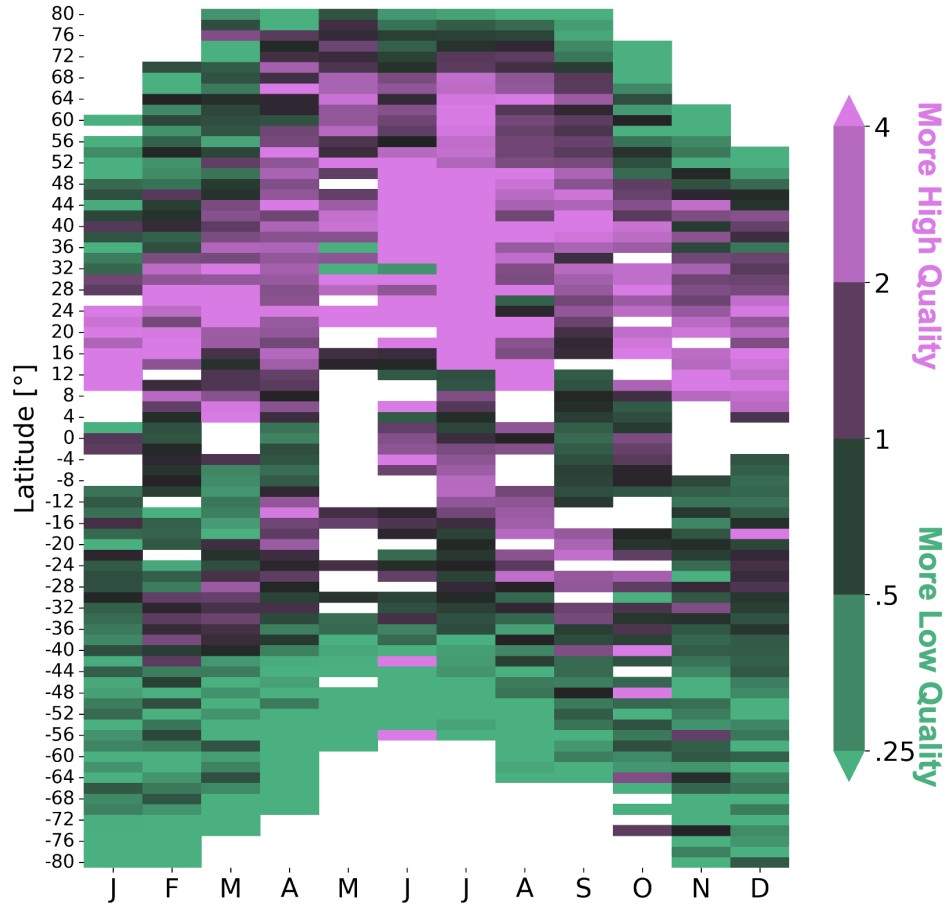

**Figure 3.** From MODIS (Aqua) L2 AODs, the ratio of 2010 MODIS QA flags of 3 (high quality) to 1 (low quality), binned by month and latitude. White areas indicate latitudes/months in which neither '1' nor '3' QC flagged retrievals were available.

Network (MAN) AODs show that .63 to .78 of SeaWiFS ocean retreivals fall within expected errors, depending on collocation selection criteria used (Sayer et al., 2012a). When validated against AERONET, land-based retrievals shows considerable regional heterogeneity, with the fraction of retrievals falling within expected errors ranging from .46 (Southern Africa) to .84 (Eastern North America); however, land regions overlapping with the (sub-)Arctic region all showed acceptable performance ($\geq$ .72 falling within expected errors) (Sayer et al., 2012b).

## 2.7 VIIRS data

The Visible Infrared Imaging Radiometer Suite (VIIRS) aboard Suomi National Polar-orbiting Partnership (Suomi NPP) is the most-recently launched instrument in this analysis, with datasets beginning in late 2011 (Wolfe et al., 2013). We use the Deep Blue L3 AOT product at 550 nm (AERDB_M3_VIIRS_SNPP, 1x1°, Aerosol_Optical_Thickness_550_Land_Ocean_Mean).



Relative to MODIS, VIIRS has fewer (36 vs. 22) spectral bands, finer maximum horizontal resolution, and the AERDB product enjoys slightly (R=.848 vs. R=.811) improved correlations with AERONET, globally (Wolfe et al., 2013; Schueler et al., 2013; Li et al., 2022).

## 2.8 MERRA-2 data

The Modern-Era Retrospective Analysis for Research and Applications, version 2 MERRA-2 aerosol reanalysis product is
derived from the Goddard Earth Observing System, version 5 (GEOS-5) Earth System Model, and the Goddard Chemistry, Aerosol, Radiation, and Transport (GOCART) aerosol module, which contains aerosol emissions, chemistry, and loss processes (Randles et al., 2017). AODs from MODIS, MISR, AVHRR and AERONET products are assimilated using the Goddard Aerossol Assimilation System (GAAS) (Randles et al., 2017).

Potential sources of error in the MERRA-2 product include inaccuracies in emissions estimates, model assumptions about
the relationship between aerosol mass and optical properties (including aerosol hygroscicity), and parameterization of wet deposition (Buchard et al., 2017). Validation of MERRA-2 AODs against MAN and US-based airborne high spectral resolution LiDAR observations showed high correlations (R= .93 and R= .85, respectively) (Randles et al., 2017). However, control simulations in which satellite retrievals were not assimilated resulted in lower correlations (R=.84 and R=.81, respectively) (Randles et al., 2017). Similarly, MERRA-2 source region extinction coefficient profiles showed greater agreement with CALIOP than
the assimilation-free control run (Buchard et al., 2017). As such, it is plausible that the product may be less reliable in the Arctic, where little data is available for assimilation. Here, we use the monthly mean instantaneous single-level assimilation aerosol optical depth analysis, for AODs at 550nm (instM_2d_gas_Nx).

## 2.9 CAMSRA data

Copernicus Atmosphere Monitoring Service reanalysis product (CAMSRA) was generated from the European Centre for
Medium-Range Weather Forecasts (ECMWF), with aerosol emissions and chemistry from the Integrated Forecasting System (IFS) chemistry and aerosol module (Inness et al., 2019). The reanalysis product assimilates 550nm AODs from MODIS Terra and Aqua, and from 2003 to 2012 the Advanced Along-Track Scanning Radiometer (AATSR) (Inness et al., 2019). Here we use the CAMSRA aerosol composition product total aerosol optical depth at 550 nm.

Gueymard et al. (2020) found slightly greater RMSEs for CAMSRA versus MERRA-2 when validated against AERONET
(.126 vs. .144), globally; at polar sites, the RMSE was more than double for CAMSRA versus MERRA-2 (.105 versus .048). As MERRA-2 assimilates AERONET, but CAMSRA does not, the extent to which reductions in RMSE reflect improved aerosol processes, rather than varying sources of assimilation, is unclear; understanding reanalysis products' capabilities and limitations away from sources of assimilation can provide better insight into the products' capabilities in the Arctic, especially in months where AERONET observations and other sources of assimilation are sparse or non-existent.





## 3 Methods

### 3.1 Comparison of Gridded Arctic AOD Seasonality

We obtained the above-mentioned L3 and reanalysis AOD datasets. Wavelengths of the AOD products range from 532 to 565 nm, with the exception of the 650 nm AVHRR product. Slight differences in AOD wavelength are unlikely to contribute substantially to differences in seasonality. However, in months characterized by an abundance of small particles (such as smoke), differences between AVHRR and other products may be expected. The gridded AODs were given at different horizontal resolutions. For each comparison, the coarser CALIOP AODs were regridded to the finer passive sensor and reanalysis grids, without interpolation.

### 3.2 Collocations

From June 2006 to September 2018, Calipso and Aqua flew together in the A-train, with Calipso trailing Aqua by approximately 2 minutes. We obtained the above-mentioned MODIS and CALIOP L2 data products for 2010, as 2010 exhibited few gaps in data availability for either instrument. Due to the high-frequency, near-simultaneous retrievals from MODIS and CALIOP, we were able to amass 138,866 strictly-defined collocations over a 1-year sampling period.

Collocations were defined as instances in which both instruments returned valid (non-Nan) AODs, and in which the mid-points of retrievals from each instrument were no more than 1 km haversine distance and occurred no more than 3 minutes from one another. Collocations were selected without replacement, such that a single retrieval could not appear more than once in the analysis; in cases where multiple matches were possible, the closest possible collocated pairing was selected. Prior studies have used radii of 25-50 km and temporal windows of 30-60 minutes when comparing satellite observations to retrievals from AERONET and DRAGON sites (e.g., Levy et al., 2013; Munchak et al., 2013; Virtanen et al., 2018). The stricter criteria made possible by the A-train configuration makes it extremely likely that collocated retrievals are sampling the same airmass.

A similar fraction of the collocations were characterized by low ('1') and high ('3') MODIS QC flags—45.8% (n=63,598) and 52.8% (n=73,302), respectively—while collocations characterized by medium ('2') MODIS QC flags contributed just over 1% (n=1,966) to the sample. As MODIS retrieves only under low cloud-fraction conditions, collocated CALIOP AODs were therefore assumed to reflect clear or nearly-clear sky conditions. Because MODIS only retrieves during the day, all collocated CALIOP retrievals were processed using the daytime algorithm.

SZAs vary with latitude, season, and time of day. Ascending passes of the polar-orbiting A-train constellation cross the equator at approximately 13:30 local time, and SZAs for collocated retrievals therefore range from 8.58° to 80.04°. Due to the configuration of the MODIS swath, the viewing angle for collocated MODIS retrievals was near-nadir, ranging from .17° to 19.22°; the relatively low range for the viewing zenith angle makes this variable well-controlled relative to the SZA. Fewer than 0.5% of the valid CALIOP day and .05% of valid near-nadir (<20°) viewing angle MODIS retrievals were selected.



# 4   Results

## 4.1   Arctic AOD Seasonality

Above the Arctic circle, most passive sensors have limited wintertime data. However, reanalysis AODs in the Arctic may be influenced by assimilated passive sensors AODs just outside the region, so in this analysis we consider seasonality within a 60-72N domain, from January 2007 to December 2021. Direct comparison of the seasonality of Arctic AODs from passive and LiDAR sensors is made challenging by the fact that there is considerable spatiotemporal variability in passive sensor data availability in the high latitudes. Fig. 4 shows the spatial distribution of data availability above 60N by instrument and season. In boreal spring (MAM), AVHRR, MISR, MODIS, and VIIRS monthly mean AODs primarily capture North Atlantic AODs, while in summer (JJA) most instruments show more uniform coverage; for MISR, MODIS, and VIIRS, autumn (SON) is characterized by a considerable latitudinal gradient in data availability, such that regionally-averaged autumnal Arctic AODs will primarily reflect sub-Arctic and low-Arctic values. As a result, characterizing Arctic AOD seasonality from the heterogenous sampling of passive sensors is inherently challenging, and comparisons between active and passive sensors need to control for sampling biases.





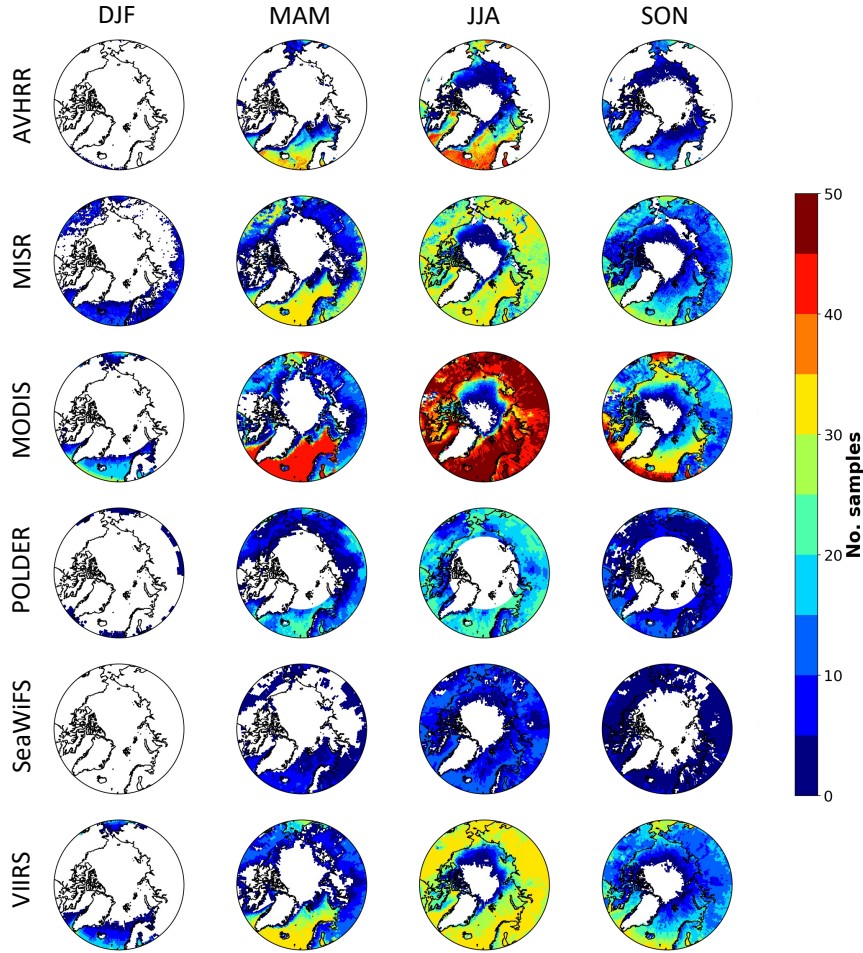

**Figure 4.** From 2007-2021, the number of valid L3 monthly mean samples above 60 N by season, for six passive sensor data products.

To control for sampling biases, for each passive sensor we subsampled the CALIOP day datasets (both cloudfree and all-sky) to only months and gridcells where the passive sensor data were available. From each passive sensor and their corresponding subset of CALIOP AODs, we calculated area-weighted monthly mean 60-72N AODs for each month, then found monthly medians from the 2007-2021 regionally-averaged AODs (Fig. 5). As this analysis is not a comparison of collocated retrievals, we also noted the number of months in the sampling period from which each monthly median value was calculated (blue numbers atop each subplot).

For MISR, MODIS, and VIIRS, there are as many or nearly as years with January and June samples. AVHRR, POLDER, and SeaWiFS do not have year-round data, but all (except AVHRR in February) have nearly as many years with samples in





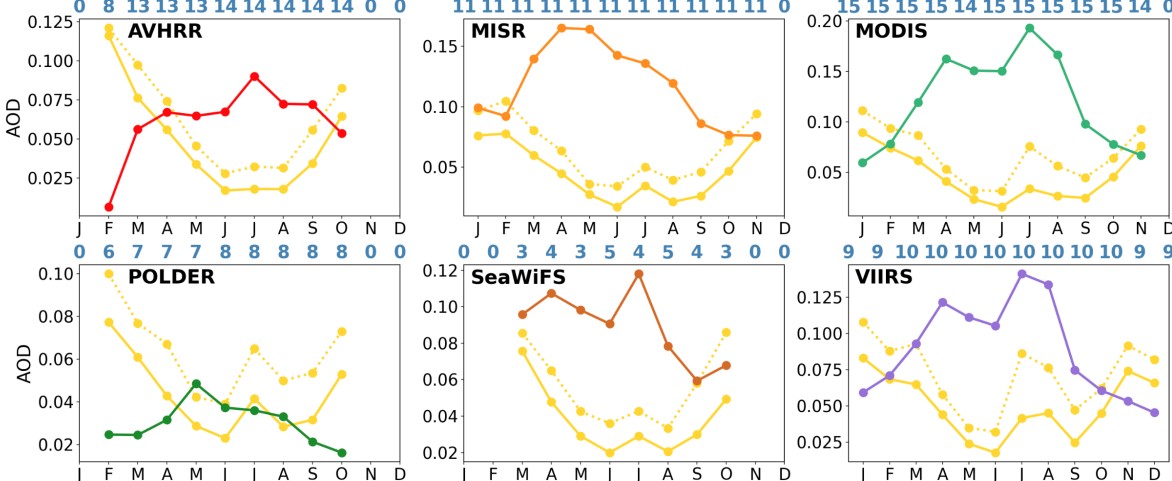

**Figure 5.** Median 60-72N area-weighted L3 AOD from six passive sensors (assorted colors) versus CALIOP (day) all-sky (solid, yellow) and clearsky (dashed, yellow) products, subsampled to only gridcells and months in which the respective passive sensor data are available. The number of months in the sampling period (7/2007-12/2021) in which at least one gridcell was available for comparison are noted in blue above each subplot, and correspond to the monthly markers on the x-axis below.

the lowest light months as in the mid-year, suggesting that median AODs are reasonable representations of each instrument's seasonality.

In each instance, pronounced differences in seasonality between CALIOP and passive sensors were evident despite controlling for sampling biases. All passive sensors (except SeaWiFS, which has no winter AODs) show substantially lower winter
versus summer AODs; for MODIS, January AODs are nearly a fourth of July values. CALIOP, in contrast, exhibits the highest AODs in winter in every subsampling, for both the all-sky and cloudfree data products.

MODIS, SeaWiFS, and VIIRS (which all make use of the Deep Blue algorithm – SeaWIFs and VIIRS exclusively) show seasonal maxima in July, with secondary peaks in April. AVHRR also shows a July peak, but with little additional seasonal structure besides an abrupt decline in AOD in February. MISR and POLDER are the only passive sensor instruments in which
AOD peaks in spring rather than summer, with instruments showing maxima in April and May, respectively. Both instruments provide additional constraints on the direction of scattering (MISR through multiple viewing angles, and POLDER through retrieval of polarized reflectance), and as a result may better control for differences between high latitude spring versus summer viewing geometries.

As passive sensors do not retrieve over clouds, the cloudfree (yellow, dashed) CALIOP data are the most apt comparison
to passive sensors. In the 65-72N comparison of CALIOP and reanalysis AOD seasonality (Fig. 1), the structure of Arctic CALIOP cloudfree AODs shows little agreement with reanalysis AODs, even in the brightest months of the year. In contrast, from April to September the structure of the subsampled 60-72N CALIOP cloudfree datasets resemble its passive sensor counterpart, particularly for MODIS, VIIRS, and SeaWiFS. Specifically, a local maximum from mid- to late summer is virtually




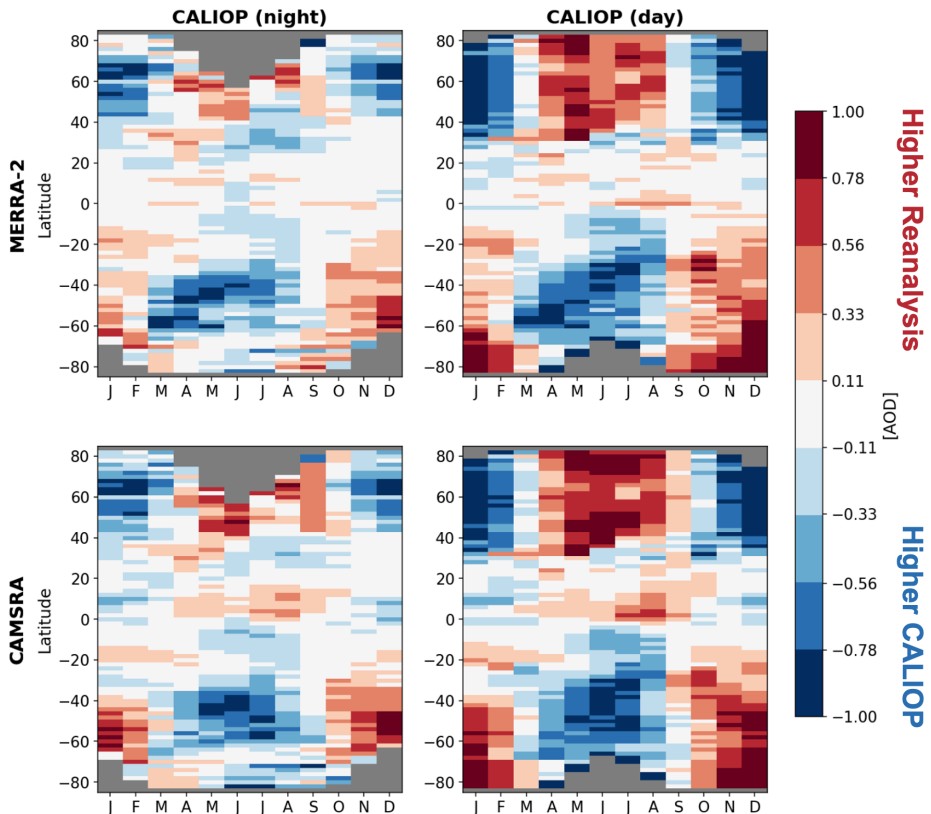

**Figure 6.** For each latitudinal band, monthly median zonal mean CALIOP (night and day) and reanalysis (MERRA-2 and CAMSRA) AODs were each normalized to their respective seasonal cycles. The difference between the normalized AODs ($\tau_{Reanalysis,normed} - \tau_{CALIOP,normed}$) with regards to latitude and month is shown, for each CALIOP (day and night) and reanalysis (MERRA-2 and CAMSRA) pair.

absent from the assessment of region-wide CALIOP monthly median AODs (Fig. 1), but is pronounced in the cloudfree datasets
subsampled to MISR, MODIS, POLDER, SeaWiFS and VIIRS data availability (Fig. 5). Such differences in the structure of the CALIOP summertime Arctic AODs when sampling from the different domains suggest that sampling biases are likely to impact region-wide assessments of summertime Arctic AODs, and likely result in an overestimate of the magnitude of the late-summer peak at the regional level. However, the decoupling of the seasonal structure from early autumn through mid-spring shows that sampling biases are not sufficient to explain overall differences in seasonality.

In the midlatitudes, reanalysis AODs are constrained by passive sensor products throughout the year, so differences in reanalysis and CALIOP seasonality likely reflects differences between passive and active sensor retrievals at these latitudes. In Fig. 5, the bias between passive and active sensor seasonality was evident down to 60N. It is yet unclear whether the bias between the seasonality of CALIOP and other gridded datasets extends beyond the high latitudes. To assess the latitudinal extent of the bias, and to separate the bias in seasonality from a bias in magnitude, for both CALIOP night and day datasets we




found the monthly median zonal mean AODs, then normalized the monthly medians to their seasonal cycle. We do the same for MERRA-2 and CAMSRA AODs, then subtract the normalized reanalysis from the normalized CALIOP AODs (Fig. 6). With this approach, we examine the bias in seasonality without regards to the magnitude of the seasonal cycle.

In each pair of reanalysis-CALIOP dataset comparisons, a bias in seasonality is evident in both hemispheres, extending from the high latitudes to approximately 40°. The bias is evident in comparisons with both the daytime and nighttime CALIOP
products, but is slightly more pronounced in comparisons with the daytime product.

These findings suggest that the mechanisms responsible for the bias in Arctic AOD seasonality between active and passive sensors may in fact be evident throughout much of the globe. Equally important, while CALIOP daytime AODs may be influenced by seasonal differences in insolation, CALIOP nighttime AODs are not. As such, differences in seasonality between reanalysis and CALIOP nighttime AODs are not driven by seasonal differences in the SNR, and another explanation for the
difference must exist. More direct comparisons between collocated retrievals is necessary to provide further information about the mechanisms and precise magnitude of the bias.

## 4.2  A Solar Zenith Angle Dependency

We hypothesized that the bias between CALIOP and passive sensor AODs may depend on the SZA, which in turn may contribute to differences in Arctic AOD seasonality. To test this hypothesis, we compare the biases between collocated CALIOP
and MODIS (Aqua) L2 AODs at different SZAs for the year of 2010.

As noted in Section 2.1, CALIOP returns AODs of zero when an entire profile is populated with RFVs. Toth et al. (2018) quantified the negative effects of all-RFV profiles on CALIOP average AODs, by comparing them to collocated MODIS retrievals. However, no studies have considered whether the frequency or magnitude of the effect of the all-RFV profiles varies with the SZA.

A-train overpasses during lower SZAs occur in the low- and mid-latitudes, making an examination of the absolute difference in AOD unhelpful–absolute differences are likely to be greatest at latitudes where baseline AODs are also highest (e.g. mid-latitude source regions). Thus, we expect that the greatest absolute differences in AOD would occur at SZAs observed during source region overpasses, regardless of any retrieval artifacts. Accordingly, for our analysis we consider a relative difference, rather than an absolute difference in AOD, where the relative difference index is defined:

$$\frac{\tau_{MODIS} - \tau_{CALIOP}}{\tau_{MODIS} + \tau_{CALIOP}}$$

This index is the difference between the instruments relative to the instrument mean, normalized to a range of $\pm 1$. For collocations in which MODIS is greater, the index will be positive, while instances in which CALIOP is greater the sign will be negative. It is yet unclear the extent to which each instrument may contribute to any observed dependencies on the SZA, and
by normalizing to the instrument-mean we do not preference the metric towards either instrument.




However, the relative difference index should only be applied to collocations in which both instruments retrieve positive values of AOD. In CALIOP all-RFV profiles (which Toth et al. [2018] found account for 71% of daytime AODs), the relative difference index will always return a value of -1 for any positive valued MODIS collocation, regardless of its magnitude. Meanwhile, negative or zero-valued MODIS AODs could result in a relative difference that is undefined, or outside the $\pm 1$ range. Accordingly, we consider two different cases:

1. the all-positives case, in which both instruments retrieve positive values of AOD

2. the zeroes case, in which one instrument retrieves an AOD of zero or less, and the other instrument retrieves some positive value of AOD

This approach has the additional benefit of allowing us to consider whether the impact of CALIOP all-RFV retrievals vary with the SZA, which can provide insight into whether this well-known mechanism by which CALIOP underreports AOD exhibits an SZA dependency. Finally, we consider whether the correlation between the collocated CALIOP and MODIS AODs depends on the SZA, which offers data users insight into the utility of different datasets under different SZA conditions.

### 4.2.1 The All-Positives Case

We found that in 105,873 of the 138,866 collocations (76.2%) identified in Section 3.2, both instruments returned positive values of AOD. Fig. 7 shows kernel density estimates (KDEs) of the relative difference index with respect to the cosine of the SZA, for all non-zero collocations (Fig. 7a), as well as for subsets in which MODIS retrievals are assigned high quality ('3') and low quality ('1') QC flags (Fig. 7b and c, respectively). A linear regression to the cosine of the SZA is also shown on each subplot. From the slope of the regression line multiplied by two, we also calculate the percent difference in the bias ($\tau_{MODIS} - \tau_{CALIOP}$) relative to the instrument-mean, that would be attributable to the SZA dependency over a theoretical $0°$ to $90°$ SZA domain (Table 1) .

Across all all-positives cases, a substantial decrease in the relative difference index with increasing SZAs is evident (Fig. 7a). The dependency would account for an approximate 97% negative difference in the bias relative to the instrument-mean, from $0°$ to $90°$ SZA (Table 1). This negative difference in the bias indicates that MODIS AODs are declining as the SZA increases, without a commensurate reduction in CALIOP AODs, or that CALIOP AODs are increasing with higher SZAs, without a similar effect on MODIS retrievals (a combination of both reduced MODIS and increased CALIOP AODs is also possible).

Subsets characterized by high and low quality MODIS retrievals (Figs. 7b and c, respectively) also show significant negative dependencies on the SZA, though the dependency for the subset characterized by low quality MODIS retrievals is substantially greater, accounting for a 132% negative difference in the bias relative to the instrument-mean from $0°$ to $90°$ SZA, as opposed to just 33% for the high quality subset (Table 1). In addition, standard errors from the calculated dependency are less for collocations characterized by low (SE = .0074) versus high (SE = .011) quality MODIS retrievals (Table 1), which is apparent in the greater concentration of the KDE around the regression line in Fig. 7c. The dependency also explains substantially more variance (R\textsuperscript2, Table 1) among the subset of low (nearly 14%) versus high (less than .5%) quality MODIS retrievals, as well as the set of all retrievals ($\approx 6\%$).





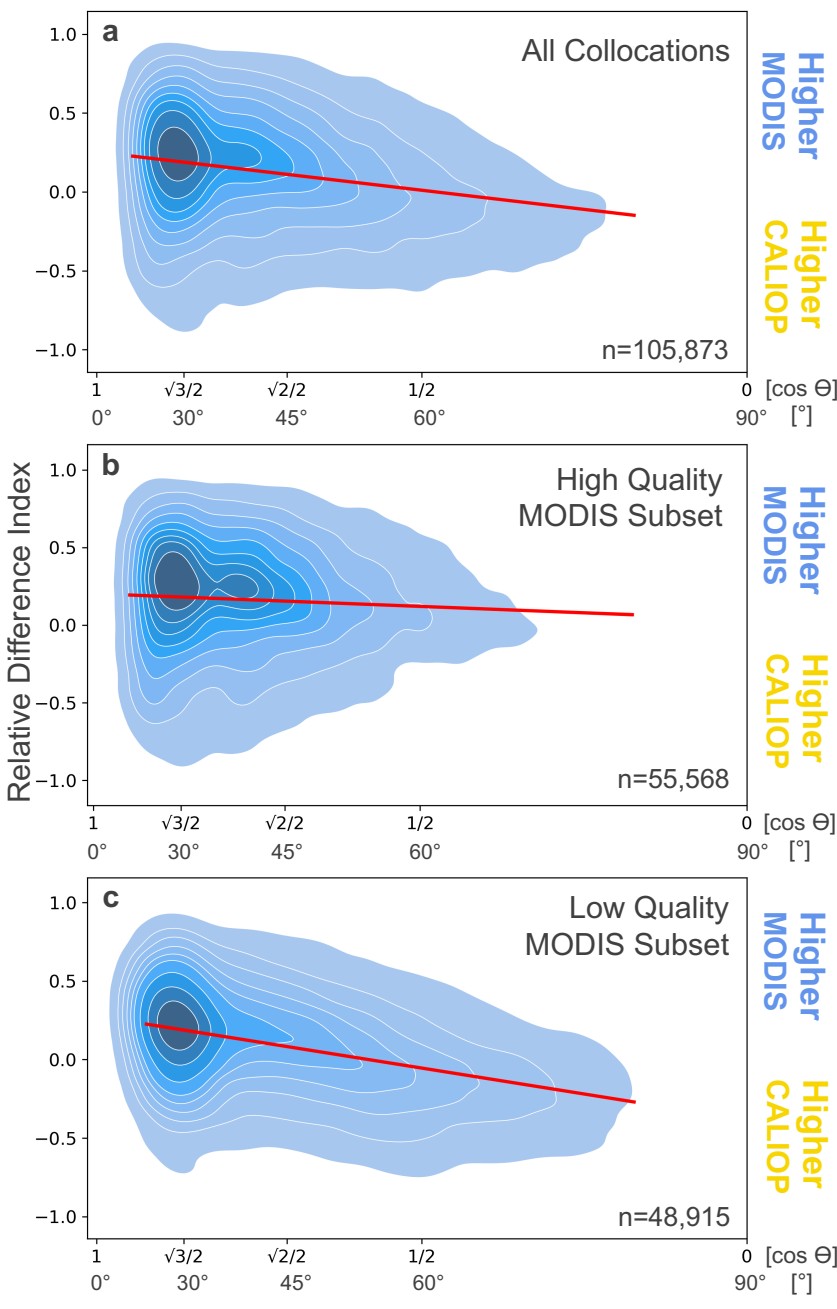

**Figure 7.** a) A kernel density estimate (KDE) of the relative difference [(MODIS-CALIOP)/(MODIS+CALIOP)] between all non-zero MODIS and CALIOP retrievals as a function of the cosine of the SZA; a linear regression to the cosine of the SZA is also shown (red). b) Same as a, except for only collocations with high ('3') MODIS QC flags. c) Same as a, except for only collocations with low ('1') MODIS QC flags.





|  | All | High QC | Low QC |
|---|---|---|---|
| m | -.48619 | -.16467 | -.66154 |
| p | 0 | $5.863 \times 10^{-53}$ | 0 |
| $R^2$ | .05670 | .00421 | .13828 |
| SE | .00627 | .01074 | .00747 |
| n | 105,873 | 55,568 | 48,915 |
| % $\Delta$ | -97% | -33% | -132% |

**Table 1.** The slope (m), p-value (p), variance explained ($R^2$), standard error (SE), number of collocations (n), and percent difference in AOD (relative to the instrument mean) over a theoretical $0°$ to $90°$ SZA domain for linear regressions corresponding to KDEs shown in Fig. 7. Values are shown for all all-positive collocations (Fig. 7a), and for subsets characterized by high and low MODIS QC flags (Figs 7b and 7c, respectively). Indications of p=0 occur when p-values were less than could be calculated with a 64-bit float.

In our set of all-positive collocations, most (85%) occurred over the ocean. No over-land MODIS retrievals were assigned a low quality flag, and few occurred at > $60°$ SZA (Fig. S1). Collocations with high quality MODIS retrievals that occurred over the ocean (Fig. S2) showed slight negative dependencies with the SZA, while those that occurred over land were slightly positive, though in both cases the dependency was low in magnitude, and with much greater standard errors, relative to what we observe in the low quality MODIS subset. The low quality MODIS subset (Fig. 7c), is therefore characterized exclusively by retrievals over the ocean. That the dependency is substantially more robust within this subset of collocations indicates that MODIS retrieval quality plays a considerable role in mediating the observed dependency. Conversely, the relatively minimal effect in the subset with high quality MODIS retrievals suggests that both CALIOP and high quality MODIS retrievals are affected relatively little by the SZA, and points instead to the low quality MODIS retrievals being primarily sensitive to changes in the SZA.

The KDE corresponding to the set of all all-positive collocations (Fig. 7a) contains roughly equal contributions from the low (48,915) and high (55,568) quality MODIS retrievals. However, as seen in Fig. 3, MODIS retrieval quality shows distinct patterns of spatiotemporal variability, with lower (higher) quality retrievals more prevalent in months and latitudes characterized by higher (lower) SZAs. Thus, the KDE corresponding to the set of all collocations (Fig. 7a) better resembles the high quality MODIS subset (Fig. 7b) at lower SZAs, and the low quality MODIS subset (Fig. 7c) at higher SZAs.

In the low quality MODIS subset (Fig. 7c), the dependency is apparent even at low SZAs. This finding suggests that the mechanism by which the dependency occurs is not simply an artifact of increasingly poor data quality at very high SZAs. Rather, it is evident at all SZAs when data quality is low, but masked by the higher fraction of high quality retrievals at low SZAs. Thus, in the L3 gridded product, averaging schemes likely dampen the effect of the dependency until the fraction of high quality retrievals is substantially diminished. Like other passive sensors data sets, the MODIS L3 AOD product shows divergent seasonality from CALIOP in the high latitudes (>60N) (Fig. 5). Such findings may plausibly be explained by seasonal differences in MODIS data quality (Fig. 3) in the (sub-)Arctic, combined with systematic reductions in AOD values at high SZAs among low quality MODIS retrievals.





### 4.2.2 The zeroes case

In 25,226 of the 138,866 collocations (18%), one instrument retrieved an AOD of zero (or less) and another retrieved a positive value of AOD. A tendency for the CAD in CALIOP processing to classify optically thin aerosol layers as clean air drives the low bias in CALIOP AODs relative to MODIS, globally; a lower SNR for CALIOP retrievals during the day results in an enhancement of this bias (Kim et al., 2017; Toth et al., 2018). Thus, one hypothetical mechanism by which active versus passive sensor AOD biases may depend on the SZA, is that the active sensor SNR may decrease with higher SZAs, resulting in a reduction in the incidence of "undetected layers," and, accordingly, all-RFV profiles. To date, no studies have examined the frequency of CALIOP all-RFV profiles against the SZA. Cases in which CALIOP retrieves an AOD of zero, but MODIS retrieves a positive value of AOD, indicate that the column is subject to one or more such undetected layers. By examining the frequency of such cases, we can also determine whether this well-known source of bias depends on the SZA.

Accordingly, we define the "zeroes case" as collocations in which one instrument retrieves an AOD of zero or less, while the other instrument retrieves some positive value. For brevity, we refer to instances in which CALIOP retrieves an AOD of zero, and MODIS retrieves a positive value of AOD, as the "CALIOP-zero" case, and instances in which MODIS returns a value of zero or less and CALIOP returns a positive value as the "MODIS-zero" case. CALIOP-zero cases will contribute to relatively higher average MODIS AODs, while MODIS-zero cases will contribute towards relatively higher average CALIOP AODs.

Fig. 8 shows the fraction of collocations within 5° SZA bins characterized by MODIS-zero (yellow bars) and CALIOP-zero (blue bars) cases; the MODIS-zeroes fractions are assigned a positive value, while the CALIOP-zeroes fractions are assigned a negative value, in accordance with the sign of their contributions to the bias (defined as $\tau_{MODIS} - \tau_{CALIOP}$) (left y-axis). To assess the contribution of each case to SZA-averaged AODs, we calculate the mean difference with the collocated positive-valued AODs within each bin and for each instrument. We then multiply these values by the corresponding fraction of zeroes-biased retrievals, for both the MODIS- and CALIOP-zeroes case; these calculations give the contribution of each case to the bias, while their sum gives the effect of all zeroes cases in total, within each SZA bin.

For SZAs bins between 20 and 80°, the CALIOP-zeroes case occurs in 9 to 23% of collocations, a substantial fraction of the total number of collocations. However, the incidence of the CALIOP-zeroes case shows no monotonic relationship with the SZA. In contrast, from 20-50° SZA MODIS-zero case consistently makes up fewer than 2% of collocations, again with no mononotic relationship to the SZA. However, beginning at 50-55° SZA, the incidence of the MODIS-zero case rises exponentially, from 1.7% at 50-55° to 13% at 75-80° SZA. The exceptionally high (67%) fraction of MODIS-zero cases in the 80°-85° SZA bin was calculated from only three total collocations, all of which were retrieved at less than 81° SZA.

These findings indicate that MODIS is susceptible to falsely retrieving AODs of zero or less as the SZA increases above moderately high (50°-55°) angles. CALIOP is also susceptible to falsely retrieving AODs of zero, and does so more frequently than MODIS in all SZA bins below 80°. However, the frequency of the CALIOP-zeroes case does not depend on the SZA, while the MODIS-zero case does.

From 20°-80° SZAs, CALIOP-zero cases consistently contribute between .0135 and .0208 to the bias, which is slightly less than the bias from undetected layers (.031 ±.052) found by Kim et al., (2017). As our analysis considers only the contribution



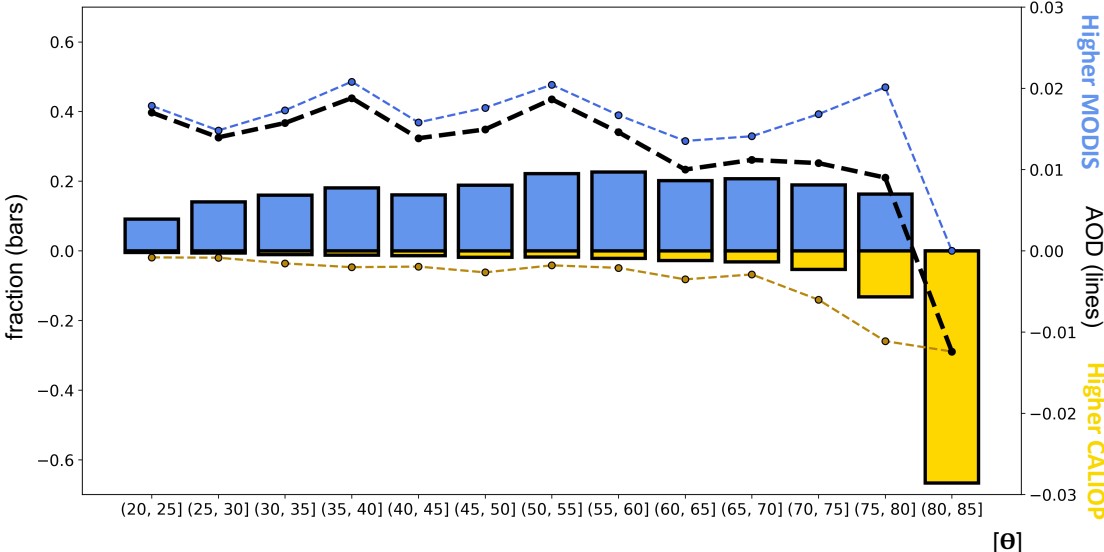

**Figure 8.** The fraction (left y-axis) of collocations corresponding to CALIOP-zero (blue bars) or MODIS-zero (yellow bars) cases, and the contribution (right y-axis) of MODIS-zero (yellow, dashed line) and CALIOP-zero (blue, dashed line) cases to the bias within each 5° solar zenith angle bin. The sum of the contributions from both instruments (black, dashed line) is shown in black, and shows the total contribution from all zeroes cases to the bias between instruments within each 5° solar zenith angle bin. .

of undetected layers resulting in a total column AOD of zero, we expect the CALIOP-zero contribution to be slightly less than the (Kim et al., 2017) estimate.

Below 55° SZA, the impact of MODIS-zero cases is small when compared to the contributions from the CALIOP-zero cases (Fig. 8). From 60-65° SZA, the MODIS-zero cases contributes -.00352 to the bias, offsetting 26% of the contribution (.0135) from the CALIOP-zero cases; from 70-75° SZA the MODIS-zero case offsets more than a third (36%) of the bias from the CALIOP-zero case, while from 75-80° SZA it offsets 55%. Thus, as the SZA increases above 50°-55°, the MODIS-zero case increasingly offsets the effects of the CALIOP-zero case. Our analysis indicates that the frequency of CALIOP all-RFV profiles do not vary substantially with the SZA, but conditions leading to MODIS retrievals of zero or less do. Thus, differences in the frequency of undetected layers likely contribute little to observed differences in CALIOP versus MODIS Arctic AOD seasonality; instead, MODIS increasingly fails to detect positive AOD values as the the SZA increases over 50°.

As mentioned in section 2.4, some negative MODIS AODs are expected when AODs are low, since negative values fall within an expected uncertainty envelope when aerosol burdens are low. A simple increase in the frequency of negative or zero valued retrievals would be expected at locations and times where aerosol burdens are characteristically low, and these retrievals may coincide with higher SZAs. However, "MODIS zero" cases as defined above are only identified when CALIOP (which is known to 'miss' optically thin AODs) returns positive values. Thus, it seems likely that the MODIS is reporting artificially low



AODs at high SZAs when column aerosol is non-negligible, consistent with our findings in 4.2.1; data quality likely similarly intervenes here.

In section 4.2.1, we found that a pronounced SZA dependency was evident in collocations characterized by low-quality MODIS retrievals, but that collocations characterized by high quality MODIS retrievals showed minimal effects. In this section, we find that instances in which MODIS retrieves and AOD of zero or less increase with the SZA, whereas CALIOP all-RFV profiles do not. Together, these results demonstrate that low quality MODIS retrievals, which are more prevalent in latitudes and months characterized by high SZAs (Fig. 3), likely drive most of the observed dependency. Still, a more quantitative measure of MODIS retrieval quality at different SZAs is necessary to understand the effects of the dependency on L3 average AODs in the high latitudes. Moreover, for many data users, the usefulness of AOD measures depends less on dataset biases, and more on whether datasets detect various aerosol events. While we have determined that the bias between datasets may depend on the SZA, it remains undetermined whether the correlation between CALIOP and MODIS AODs varies with the SZA.

### 4.2.3 Usefulness of low quality retrievals

In remote regions where observations are sparse, extracting as much information as possible from all available data sources is compelling. In many cases, imperfect or incomplete data can provide useful information, but only when the biases and limitations of the data are understood.

Validation of AOD data products involves assessing both the bias and the correlation between instruments. Bias provides important context for interpreting differences between the instrument retrievals, while correlation determines whether the data capture similar phenomena. Above, we examined whether the bias between CALIOP and MODIS products depends on the SZA. Now, we consider whether the correlation shows similar dependencies.

To determine whether the SZA affects the correlation between CALIOP and MODIS, we calculated Pearson's R for the above collocations within running 5° SZA bins (Fig. 9, black line). To examine the extent to which retrieval quality and hemispheric location intervene in this relationship, we performed the same analysis for subsets of collocations characterized by high- (purple line) and low- (green line) quality MODIS retrievals, and separately for subsets from the NH (blue dashed line) and SH (blue dotted line). The fraction of collocations within each 5° SZA bin characterized by low quality MODIS retrievals is shaded in gray (left y axis).

The correlations for all collocations (black line) shows a pronounced negative SZA dependency, beginning at even low ([22°, 27°)) SZAs. A brief departure from the overall negative relationship begins at [56°, 60°) and peaks at [63°, 68°) SZA, with R nearly doubling from .340 to .570. The departure is followed by a precipitous drop in correlation at increasingly high (> [66, 70°)) SZAs. This temporary increase in correlation is evident in collocation subsets characterized by NH collocations and high-quality MODIS retrievals, but not SH collocations or low-quality MODIS retrievals. Retrievals with SZAs between 58° and 65° in the NH correspond to midlatitude source regions, and enhanced correlation likely stems from higher average AODs (and hence greater variability), and perhaps better constraints on retrieval processing in more accessible regions.

Across all SZAs, the subset of collocations with high-quality MODIS retrievals shows no monotonic relationship to the SZA. For this subset, R remains consistently high at all SZAs, ranging from .512 to .828. In contrast, the subset characterized





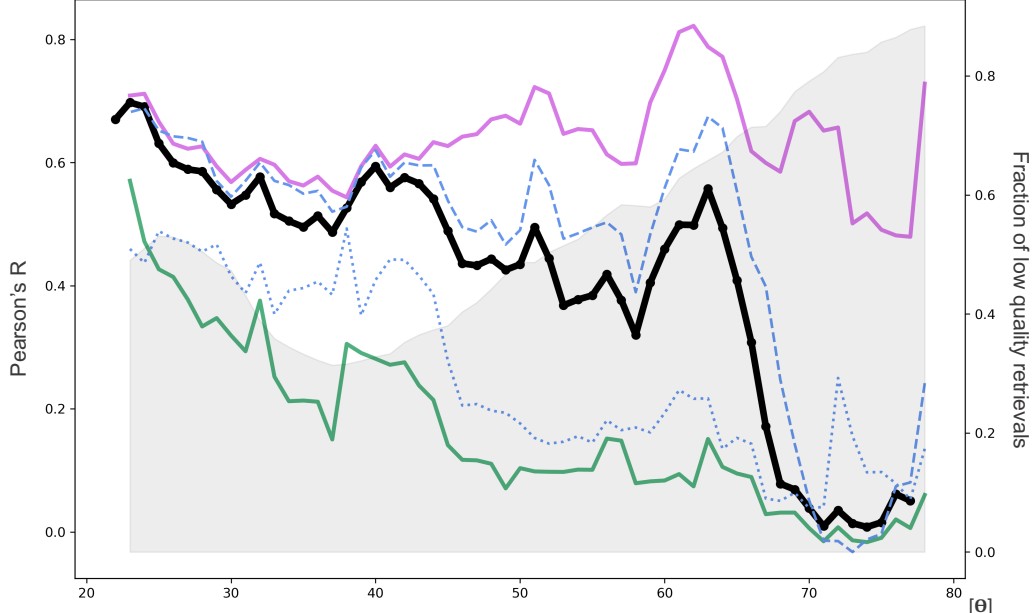

**Figure 9.** Pearson's R for collocated MODIS and CALIOP AODs within running 5° SZA bins (left y-axis), for all collocations (black line), collocations with high quality MODIS retrievals (purple solid line), collocations with low quality MODIS retrievals (green solid line), NH collocations (dashed blue line), and SH collocations (blue dotted line). The fraction of collocations within each running 5° SZA bin characterized by low quality MODIS retrievals is shown by the gray shading (right y-axis).

by low-quality MODIS retrievals shows lower correlations than the high-quality subset at virtually all SZAs, as well as a pronounced, steady decline in correlation with increasing SZAs. At low (<[22°, 26°)) SZAs, low quality MODIS retrievals are suitably correlated with their CALIOP counterparts (R>.5). Above moderate SZAs [>45°, 49°), the correlation never surpasses R=.3, while above [75°, 79°) SZA the correlation effectively disappears as R never exceeds .1.

From >[35°, 39°) SZA, the fraction of collocations characterized by low-quality MODIS retrievals increases steadily with
higher SZAs. Accordingly, the decline in correlation among all collocations is more pronounced than for just the low-quality MODIS retrieval subset, reflecting both the SZA dependency for low-quality MODIS retrievals, and the increased fraction of low-quality retrievals with higher SZAs. Similar to our findings in section 4.2.1, the decline in correlation does not simply result from a decline in retrieval quality alone; it is evident at all SZAs among low quality retrievals, and the increased incidence of low-quality retrievals at higher SZAs amplifies the relationship when averaging across all collocations.

The decline in correlation is evident in collocations from both the NH and SH, though SH collocations (blue, dotted line) are less correlated than NH correlations (blue, dashed line) at all SZAs below 70°, and do not show the abrupt decoupling at [63°,68°). Instead, a more modest decline is observed after [42°, 47°) SZA. SH retrievals with SZAs of 45° to 60° occur most frequently over the Southern Ocean, where MODIS retrievals are characteristically low-quality (Fig. 3). Accordingly, hemispheric differences in the relationship between correlation and the SZA can be explained by slight differences in the





spatiotemporal variability of data quality. In both hemispheres, CALIOP and MODIS low-quality retrievals gradually decouple as the SZA increases, and the prevalence of low-quality MODIS retrievals intervenes in the relationship between correlation and SZA.

At SZAs greater than 60°, low-quality MODIS retrievals make up 60% or more of the retrievals, while at very high (>70°) SZAs, MODIS retrievals are overwhelmingly (>80%) low-quality; thus, at high SZAs even quality-weighted MODIS L3 prod-
ucts rely heavily on low quality retrievals. Accordingly, L3 products' representations of high latitude spring and autumn AODs are likely influenced by the dependencies noted in sections 4.2.1 and 4.2.2.

The low correlation between CALIOP and MODIS retrievals at high SZAs indicates that the L3 data products become increasingly less likely to capture similar aerosol events at high SZAs, even when the effects of the SZA dependency of the bias are removed. Had the bias, but not correlation, shown an SZA dependency, data products in the high latitudes may be
relied upon to show similar sub-seasonal variability, and similar interannual variability for a given month, even while the seasonality diverged. However, our findings indicate that the MODIS L3 products have limited utility when a preponderance of the gridcell-averaged L2 retrievals occur at SZAs above [63°,68°) SZA. However, where sufficient coverage with high-quality L2 MODIS AODs is available, such retrievals may provide useful information even under very high (>70°) SZAs.

## 4.3 Summary and Conclusion

The Arctic is characterized by both heightened sensitivity to radiative forcing and a paucity of observational constraints. As such, satellite data products have the potential to provide valuable information about the region, but only if their capabilities and limitations in the region are understood and, ideally, quantified. In this analysis, we found stark differences in Arctic AOD seasonality between CALIOP and six passive sensor L3 data products, even when controlling for sampling biases, with all passive sensor products reporting dubiously low AODs from early autumn to mid spring. We further found that biases in
seasonality between CALIOP and two reanalysis products were evident in both hemispheres, and extended from the poles into the midlatitudes. As the reanalysis products assimilate passive sensor AODs, this suggests that biases between passive and active sensor AOD seasonality may be evident in the midlatitudes, and may be propagated into reanalysis products.

Our analysis of collocated L2 MODIS and CALIOP retrievals demonstrates that the bias between the instruments is susceptible to a substantial and significant dependency on the SZA, and that this dependency is mediated by an increase in the
frequency of low-quality MODIS retrievals at higher SZAs. The dependency can be observed in both the relative reduction in the magnitude of positive MODIS AODs compared to CALIOP, as well as the increase in instances of zero or negative valued MODIS AODs under conditions in which CALIOP AODs are positive. We further find that the correlation between CALIOP and MODIS AODs decreases as SZA increases, and that this decoupling of the data products is similarly mediated by the increased fraction of low quality retrievals at higher SZAs.

Biases related to undetected layers and, hence, RFVs in the CALIOP daytime AOD products are well-documented, and conceivably could vary with the SZA. However, our analysis found that the incidence of all-RFV profiles do not show an SZA dependency (Fig. 8). Moreover, the absence of a robust dependency in the high-quality MODIS retrieval subset (Fig.7) suggests that neither CALIOP nor high-quality MODIS AODs show substantial dependencies on the SZA, and thus points to




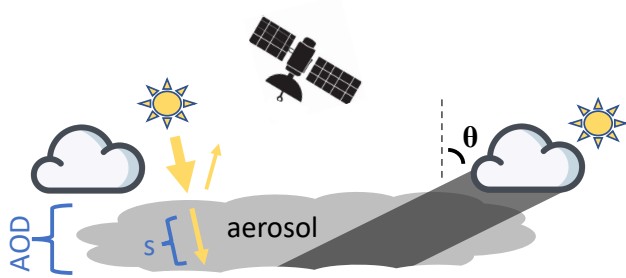

**Figure 10.** A schematic illustration of insolation and reflectance at low (left) and high (right) solar zenith angles (θ), with nearby clouds. At high SZAs, adjacent clouds may enshadow the aerosol layer, resulting in lower reflectance and underestimates of AOD for passive sensors. At low SZAs, the light source is unimpeded by neighboring clouds.

the low-quality MODIS retrievals as the predominant source of the dependency. It is therefore unlikely that errors in CALIOP
LiDAR ratios drive the dependency on a global scale. Collocations with low-quality MODIS retrievals occurred only over the
ocean, where MODIS radiative transfer assumes a pseudospherical atmosphere; accordingly, plane parallel assumptions do not
explain the robust SZA dependency observed in this subset of collocations.

The mechanism by which low-quality MODIS retrievals return lower values with higher SZAs is worthy of further investigation. QC flags of 1 are frequently applied to scenes characterized by higher cloud fractions, ocean glint, or nearby snow and
ice. The inversion algorithm for MODIS calculates aerosol reflectance from top of atmosphere (TOA) reflectance, as well as
assumptions about background noise. Aerosol reflectance, and hence AOD, may be underestimated when TOA reflectance is
under reported (such as when an aerosol layer is shadowed by a neighboring cloud, as shown in Fig. 10), or when background
noise, such as surface reflectance, is over reported (e.g. when the surface is enshadowed by a cloud). Further comparisons of
satellite AODs to AERONET, MAN, and ground-based LiDAR at different SZAs are will help disentangle these effects, and
should be the subject of future investigations.

This analysis considers only the effects of the SZA on retrievals with low viewing zenith angles, yet Level 3 products are
averages of retrievals with a range of viewing angles. It is reasonable to expect that different viewing geometries may alter
the effect of the SZA on low quality MODIS retrieval biases, and quantifying the aggregate effect of the observed dependency
on Arctic AODs in L3 products requires more constraints on the effects of complex viewing geometries. Further, sources of
error mentioned above (unaccounted for seasonal differences in the LiDAR ratio, plane-parallel assumptions over high latitude
land) that are not evident in the global analysis may still affect retrievals on more local scales. Still, our findings show that
MODIS low quality retrievals, which become predominant in the (sub-)Arctic from September to April, decline substantially
and systematically with higher SZAs, consistent with the improbable decline in L3 passive sensor AODs over the same months.
The identified mechanism likely shapes high- and even mid-latitude seasonality in passive sensor L3 products.



Finally, high quality L2 passive sensor retrievals may provide an appealing alternative for studying Arctic AOD seasonality, though only when sampling biases are well-controlled. Efforts to better understand and characterize variability in Arctic aerosol can help contextualize results from high quality passive sensor L2 retrievals in the Arctic, and are necessary for understanding aerosol processes in the region.

*Author contributions.* SS developed the hypothesis, methodology, and performed the investigation of this analysis, with feedback and suggestions from YW and MT and RL. YW and MT helped conceptualize methods for controlling for sampling biases in results section 4.1, while RL provided feedback on data products, suggested partitioning collocations by QC flag in section 4.2, and provided feedback on explanatory mechanisms to describe the observed findings. SS wrote the original manuscript, with edits from RL, YW, and MT

*Competing interests.* The authors declare that they have no conflict of interest.

*Acknowledgements.* This research has been supported by the National Science Foundation Office of Polar Programs (grant no. OPP-1825858), the National Aeronautics and Space Administration (award no. 80NSSC20K1254), and Earth Science Senior Review: Terra and Aqua Algorithm Maintenance.

*Data availability.* L3 products from MODIS (doi: 10.5067/MODIS/MYD04_L2.061), CALIOP (doi: 10.5067/CALIOP/CALIPSO/CAL_LID_L3_Tropospheric_APro_AllSky- Standard-V4-20), and VIIRS (doi: 10.5067/VIIRS/AERDB_M3_VIIRS_SNPP.011), and L2 products from CALIOP (doi: 10.5067/CALIOP/CALIPSO/LID_L2_05KMAPRO-STANDARD-V4-20) and MODIS (doi: 10.5067/MODIS/MYD04_L2.061) can be downloaded from NASA's Atmospheric Science Data Center at Langley Research Center, while the SeaWiFS data product (doi: 10.5067/MEASURES/SWDB/DATA304) is stored at NASA Goddard Space Flight Center. POLDER (https://download.grasp-sas.com/download/polder/polder-3/models/v2.1/l3/1_degree/monthly/) can be accessed through AERIS/ICARE Data and Services Center. AVHRR (doi: 10.7289/V5X9287S) can be accessed through the National Centers for Environmental Information (NCEI) at NOAA. MERRA2 (doi: 10.5067/2E096JV59PK7) can be accessed through the NASA Global Modeling and Assimilation Office. CAMS reanalysis product (Inness et al. (2019), http://www.atmos-chem-phys.net/19/3515/2019) are available from the Copernicus Atmospheric data store.



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
