# Peer review of "The Solar Zenith Angle Impacts MODIS versus CALIPSO AOD Retrieval Biases, with Implications for Arctic Aerosol Seasonality"

_EGUsphere, 2024_

## Author Comment (AC1)

**Review 1**

The paper demonstrates a difference between the seasonality of Arctic aerosol loading reported by passive imagery and orbital lidar/surface observations. Lidar and reference observations see maximal aerosol optical depth during winter, with two to four times greater values than during summer. A suite of six passively sensed datasets exhibits the opposite behaviour. The cause of this is explored for the combined Dark Target Deep Blue (DTDB) MODIS product by examining the relative difference between it and CALIOP observations at Level 3. It is shown that the relative difference becomes increasingly negative as solar zenith angle (SZA) increases and that this is concentrated within MODIS observations flagged as lower quality. Some possible explanations for this trend are eliminated, such as CALIOP's sensitivity changing with SZA. The authors argue that the erroneous seasonality is caused (at least in part) by large SZA during winter resulting in a preponderance of low-quality MODIS retrievals, which have a lower sensitivity to aerosol (and/or systematically underreport AOD) and, therefore, improperly reduce L3 AOD over the Arctic during winter.

I recommend this paper for publication after considering some minor points. It was an engaging and interesting read. I think I was already aware of some of the central points – MODIS retrievals are less accurate at large SZA and Arctic variability is poorly captured – but this manuscript is a thorough examination of the topic and is more accessible than the technical reports where the information is currently presented. A selection of minor comments and technical corrections follow for the authors to consider in the event that their submission is revised.

**Thank you for your feedback here and below. Your comments were helpful and offer useful opportunities to strengthen the manuscript. One point of clarification that has arisen in both reviews relates to this statement:**

*The authors argue that the erroneous seasonality is caused (at least in part) by large SZA during winter resulting in a preponderance of low-quality MODIS retrievals, which have a lower sensitivity to aerosol (and/or systematically underreport AOD) and, therefore, improperly reduce L3 AOD over the Arctic during winter.*

**We argue that retrieval quality (which is not explicitly determined by the SZA, despite higher SZAs coinciding with increased incidence of low-quality retrievals) affects the sensitivity of retrievals to increases in the SZA, such that when retrieval quality is low (as in the Arctic winter), AODs will decline with the SZA. In accordance with Reviewer 2's recommendation that we change the title to better reflect this point, we updated the title to read:**

"Seasonality Biases Arise from the Interplay of Retrieval Quality and Solar Zenith Angle Effects in Passive Sensor AOD Products"

**We also updated parts of the conclusion (lines 568-604) to ensure that this is clear—these and other substantive changes to the manuscript are marked in red. From both reviewers, it was also evident that more detailed discussion of the QA-flag assignment is necessary, and will help readers better understand the relationship between quality and the SZA, which we explain in more depth below.**

L379: The paper would benefit from a more detailed explanation of what the QA flags denote and how they are derived. It would alleviate my concerns over the absence of low-quality observations over land, which is unexpected as Arctic land is one of the most difficult environments over which to retrieve aerosol. Further, the supplementary figures imply that the main document should only discuss retrievals over sea as the land appears to be virtually unaffected by SZA. Given Rob is on this paper, I'm sure the authors properly understand the quality flags but it may be helpful to briefly explain their derivation. (A flowchart would be lovely as my experience is that MODIS QA is described over several documents, some of which amend previous versions.) My reading is that DB QA is based on variance in the pixels, while Tables C1 and 2 of Levy et al 2016 state that there is only one way for land pixels to be flagged QA=1 (having between 21 and 30 pixels) while ocean has several routes, such that QA doesn't have a consistent meaning between the two domains. Personally, I'd have dug into the QA bitmasks to see if the SZA effect was constrained to specific channels or surface conditions but that is too much work for a correction.

> I also note point F on page 3 of https://atmosphere-imager.gsfc.nasa.gov/sites/default/files/ModAtmo/Collection_006_Changes_Aerosol_v28.pdf, which seems relevant to the zeroing of AODs over ocean discussed in this paper.

**To address both this question and comments from the other reviewer, we added a new subsection on "Quality Flags" to the section on MODIS data section. This section now also describes how the "combined DBDT" product screens out low-quality retrievals over land. We also added a table to the supplement outlining the specific thresholds involved in QAC flag assignment, which does indeed highlight the different pathways for ocean versus land retrievals. It's worth noting that DB retrievals over land only allow QAC scores of 0, 2, and 3.**

**We especially appreciate the reviewer's suggestion about using run flags/bitmasks, and think this would be an important next step in understanding the specific mechanisms driving the dependency; we added a section to the conclusion discussing this as an important next step. To be consistent with the most recent documentation, we also updated "QC flags" to "QAC flags" throughout the manuscript. Below is the section on Quality flags that we added to the manuscript:**

**2.4.2 Quality Flags**

Both the DB and DT algorithms operate on the same NxN boxes of native-resolution pixels, with similarities and differences in their logic and selection of appropriate wavelength band channels.  Each algorithm performs similar types of data filtering, including cloud masking, snow masking, and other forms of pixel screening; however they are performed independently defined according to algorithm-specific criteria.  For DT, the remaining valid pixels are sorted so that a portion of the brightest (and darkest) pixels are discarded to reduce the influence of outliers; if a sufficient number of valid pixels remain, the algorithm computes a single representative top-of-atmosphere spectral reflectance vector (Levy et al., 2024). For DB, the spectral reflectance vector for each unscreened pixel is retained (Hsu et al., 2013). Both algorithms then perform inversions by comparing these observed reflectances to values in look-up tables (LUTs), which provide expected reflectances under a range of retrieval scenarios. The AOD corresponding to the closest match between observed and LUT-based reflectances is selected as the retrieved value. Finally, while DT has already operated on a single vector to retrieve a single retrieval result in the NxN box, DB averages the multiple retrievals to provide a final result for that NxN box.

During the process for either algorithm, various indicators may arise suggesting that the retrieval may be more or less robust. These indicators are then used to assign to the retrieval a Quality Assurance and Control (QAC) flag, which conveys the algorithm's assessment of retrieval reliability. QAC values range from 0 ('no confidence') to 3 ('high confidence' or 'high quality'). To receive a QAC flag of '3,' several retrieval conditions must be met, depending on the algorithm and whether the retrieval occurs over land or ocean. Over ocean, high-quality MODIS retrievals require successful use of all relevant spectral channels (including 1.65 μm and 2.13 μm), low reflectance variability across the retrieval window, low fitting errors with LUTs, and minimal contamination from ocean glint (Hubanks, 2021). Over land, DT and DB QAC scores depend on the number of pixels remaining within a scene after masking and outlier removal; DB also accounts for the standard deviation of AOD within the scene (Hsu et al., 2013; Hubanks, 2021). None of the QAC flag assignments depend explicitly on retrieval geometry, though retrieval geometry may influence the conditions used to assign QAC values. More detail on the thresholds used to assign each QAC score is provided in Table S1.

QAC flags of '1' or '2' are assigned when retrievals meet minimum quality thresholds but are affected by suboptimal conditions—such as a low number of valid pixels, moderate glint contamination, missing data in a shortwave infrared (SWIR) channel, elevated LUT fitting errors, or high reflectance or AOD variability. While these issues degrade retrieval confidence, they are not severe enough to warrant exclusion (QAC = 0). These lower-quality retrievals (QAC = 1 or 2) often reflect scene complexity or suboptimal viewing conditions, factors that increase the likelihood of 3D artifacts. For example, scenes containing clouds lead to a higher number of cloud-masked pixels, resulting in lower QAC scores. Cloud adjacency and other 3D effects can affect neighboring pixels by shadowing or

illuminating features in ways not captured by the 1-dimensional inversion. These effects represent a persistent and poorly quantified source of uncertainty in aerosol remote sensing.

L173: Are you sure the L3 averaging disregards QA? Page 10, paragraph 2 of https://atmosphere-imager.gsfc.nasa.gov/sites/default/files/ModAtmo/ATBD_MOD04_C005_rev2_0.pdf states, "Those retrievals with QAC=3 are assigned higher weights than those with QAC=2 or QAC=1." Apologies if I missed a later revision.

**Yes, the product we use (AOD_550_Dark_Target_Deep_Blue_Combined) was new with Collection 6 (https://modis-images.gsfc.nasa.gov/_docs/L3_ATBD_C6.pdf):**

**Deep Blue Aerosol   &   Deep Blue/Dark Target Combined**

| Input L2 SDS | Old C005/051 L3 SDS | New C006 L3 SDS | L3 Statistics |
|---|---|---|---|
| Deep_Blue_Spectral_Aerosol_Optical_Depth_Land | Deep_Blue_Aerosol_Optical_Depth_Land | Deep_Blue_Aerosol_Optical_Depth_Land | S, HC |
| Deep_Blue_Aerosol_Optical_Depth_550_Land | Deep_Blue_Aerosol_Optical_Depth_550_Land | Deep_Blue_Aerosol_Optical_Depth_550_Land | S, HC |
| Deep_Blue_Angstrom_Exponent_Land | Deep_Blue_Angstrom_Exponent_Land | Deep_Blue_Angstrom_Exponent_Land | S, HC |
| Deep_Blue_Spectral_Single_Scattering_Albedo_Land | Deep_Blue_Single_Scattering_Albedo_Land | Deep_Blue_Single_Scattering_Albedo_Land | S, HC |
| Deep_Blue_Number_Pixels_Used_550_Land | (not in C005/051) | Deep_Blue_Number_Pixels_Used_550_Land | S, HC |
| AOD_550_Dark_Target_Deep_Blue_Combined | (not in C005/051) | AOD_550_Dark_Target_Deep_Blue_Combined | S, HC |

**Above is a table from page 121, which shows that simple statistics (S) and histogram counts (HC), but not quality weighted (QA) statistics are available for the L3 product. The description of the L3 statistics can be found on pages on pages 22-28, with the "mean" variable used in this analysis described accordingly:**

3.2.1. Simple statistics  Mean. Statistics always have the Scientific Data Set (SDS) name suffix "_Mean" and are computed by taking an unweighted average of L2 pixels (sometimes sampled, see Table 1) within a given 1° L3 grid cell.

L465: While I agree with your point here, I feel that the problem in the L3 data considered is more about producing useful uncertainty estimates to either reintroduce weighting (if my above point is wrong) or fix the existing one. I've been to enough AEROSAT discussions to know why the DTDB team is resistant to that approach – and I'm not asking for the authors to apply it here – but I think this is a good opportunity for the authors to discuss what uncertainty information would be needed. The data presented in this study could be used to include an SZA term within the expected error envelope of low-quality data. What validation campaigns or sites would be necessary to properly understand these limitations? When bidding for new infrastructure, it would be useful to be able to point at a direct request from an independent team.

**The authors agree that this is an interesting idea and that relating retrieval geometry to expected errors could be an important next step following this project. We also note that in cases where AOD is low, the retrieved values may fall well within the expected error range yet still yield low correlation coefficients, making additional analysis necessary if we want to relate viewing geometry to the expected error. We adapted the following paragraph in the conclusion to address this possibility directly (lines 596-604):**

Finally, for low-quality retrievals we found an SZA dependence in both biases and correlations, suggesting the potential for bias correction or more detailed error characterization in future products. Inversion algorithms for MODIS depend on measured top-of-atmosphere reflectance and assumptions about surface reflectance, path reflectance, and multiple scattering. While our analysis shows that CALIOP sensitivity does not appear to vary with SZA, ground-truth measures are necessary to provide important constraints on these assumptions under different retrieval geometries. Further comparisons of satellite AODs to AERONET, MAN, and ground-based lidar across a range of SZAs can help disentangle the effects of 3D artifacts on these different assumptions, and should be a focus of future investigations. Additionally, examining low-quality retrieval errors within different SZA and VZA bins can help determine the extent to which expected errors may vary with retrieval geometry, supporting more refined uncertainty quantification in future products.

**Correlation coefficients tend to decrease at high solar zenith angles, but correlation alone does not reflect the magnitude of retrieval differences. Even when correlation is low, retrievals may still agree closely in absolute terms. This limitation highlights the need to examine expected errors directly, rather than relying solely on correlation, to better assess retrieval performance under varying viewing conditions."**

L284-287: I'm not sure I agree with the wording used here: the reasonableness with which a median represents the sampled population is determined by the distribution of the quantity measured and there isn't a single sample size that achieves that for all distributions. However, I believe what you were attempting to say is that the number of samples is basically constant through the year for each sensor and, therefore, there is no expectation that the shape of any curve in Fig. 5 has been influenced by the number of observations.

**Updated. The paragraph (lines 327-330) now reads:** "For MISR, MODIS, and VIIRS, the number of years with January samples is comparable to that with June samples. While AVHRR, POLDER, and SeaWiFS lack full year-round coverage, each (except AVHRR in February) has a similar number of sampled years in the low-light months as in mid-year. Given the relatively stable sampling across months, wintertime median AOD values are as representative of the dataset as summertime values."

."

L345: I'm also not sure I agree with this wording: if one instrument consistently reports much smaller values than the other or one has substantially larger variability, it could still 'dominate' the metric. You normalize because the values you wish to evaluate cover an order of magnitude while suffering approximately constant uncertainties, such that a relative metric is more informative of the full range than an absolute one.

**We clarified the wording in response to the comment—our aim is to explain the rationale behind using the instrument-mean as the normalization factor, as we have no ground-truth. The sentence (line 385-386) now reads:** "As neither instrument serves as a ground-truth reference, we normalize to the mean of the two instruments rather than to either individually."

L366-378: The description of the slopes here was difficult for me to understand. When you say "an approximate 97% negative difference in the bias relative to the instrument-mean, from 0 to 90 SZA", do you mean "if the red line in Fig. 7a were extended across the full range of x, then the difference between its maximal and minimal values is 0.97"? My first guess was that you meant that the ratio between the slope and the intercept was 0.97, but eventually realised that was non-sensical. My problem may have been recalling that the y-axis is a ratio that can be expressed as a percentage. A different framing may be clearer to a reader encountering this data for the first time, such as writing the slope in the form

**We agree that the phrasing is confusing. We updated this by adding more description up front (lines 404-411), and then referring to the relative difference index RDI and the %Δ throughout the remainder of the document:**

A linear regression of the form:

$RD = m * \cos(\theta) + v$

is overlaid on each subplot, where the slope (m) describes the sensitivity of RDI to the SZA (θ). As the RDI is ½ the bias divided by the mean of both instruments, multiplying the slope

by 200 (i.e., 2 × 100) yields the percent change (%Δ) in bias relative to the instrument-mean over a unit change in cos(θ)—equivalent to a shift from 0 to 90 SZA. To calculate the percent change over a specific SZA range (%$\Delta_{\theta 0 \to \theta 1}$), this value can be further multiplied by the difference in cos(θ) between the lower and upper bounds of that range, as described below:

%$\Delta_{\theta 0 \to \theta 1}$ = m*[cos(θ$_0$)-cos(θ$_1$)]*200

L486-497: I know that [75,79) means $75 \ge SZA < 79$. I do not know what <[22,26) means nor how it differs from [>45,49). My best guess is "At low (< 22) SZAs… Above moderate (>49)…$.

**We updated these to avoid confusion.**

L384: It occurs to me that it is possible to include binary variables within regression models, such that a simultaneous regression of relative difference against cos SZA and high/low quality could be done in a future study. I've never done it myself, but I have seen such regressions applied to polls using party identification and income or age as variables. On page 12, you say that the CALIOP data is subsampled to only cells where passive sensor data is available. I am personally curious how these compare to (a) each other and (b) the total population. It doesn't need to be in the final paper but, if you have the time, I would be greatly appreciate seeing a single plot of the solid yellow lines of Fig. 5 alongside the equivalent for all points in your reply. Their spread would be a simple estimate of the effect of sampling caused by cloud and failed retrievals.

**Binary regression offers an interesting option that we will consider for future work—we appreciate the suggestion. Below are figures showing the subsampled CALIOP data for both cloudfree and all sky datasets. The legend indicates the passive sensor data product to which the CALIOP daytime product was subsampled.**

[Figure]

**Especially in the late summer period, the median values appear somewhat sensitive to sampling. This may partly reflect recent trends in wildfire smoke emissions, as subsampling to VIIRS (launched in 2011, the latest of all instruments included in the analysis) resulted in the greatest late-summer values. Differences between cloudfree and all-sky days in the late summer are also evident.**

Technical corrections:

You are inconsistent in hyphenating "low-quality" when used as an adjective.
L39: The EarthCARE
L45: I think it should be 'depends' as the sentence subject is 'representation' rather than 'AODs'.
L92: dark target product assumes
L213: Aerosol
L248: non-NaN
L455 retrieves an AOD

**updated**

**Review 2**

This manuscript delves deeply into the question of why passive sensors, particularly MODIS Dark Target/Deep Blue aerosol products, do not capture the same seasonal cycle as does CALIOP. The study accounts for sampling biases, solar zenith angle, data retrieval quality flags and includes analysis of situations where one of the sensors returns a 'zero', as well as when both sensors report positive AOD. The study shows that assimilation data sets tend to follow the passive sensor seasonal signatures because they are dependent on the passive sensors, so that the results have significant consequences down the line and across disciplines. The authors do a very complete job, examining biases and correlations

between the passive and active sensors.  The figures are informative and the manuscript is very easy to read. I recommend publication.

Although I do want to point out that at the end we learn quite a bit about retrieved aerosol data sets and their biases, but not very much about Earth's atmosphere or aerosols.  I personally would have submitted this to *Atmospheric Measurement Techniques*, not to *Atmospheric Chemistry and Physics*.

 I have no need to remain anonymous.  This is Lorraine Remer writing.

There are a few things that caught my eye as I was reading.

**Thank you Dr. Remer for your thoughtful feedback. Your comments have been helpful and are answered in more detail below. You raise an important point about the journal choice, and indeed this was something we considered as well. We chose ACP because the paper relates aerosol seasonality—a topic of interest to data users—to retrieval quality and geometry—areas often discussed among developers but less frequently explored from a user-oriented perspective. Central to our discussion is how retrieval biases shape interpretations of aerosol seasonality. This focus aligns with ACP's scope, which includes "studies with important implications for our understanding of the state and behaviour of the atmosphere and climate." The feedback from Initial Evaluation #2 emphasized that the paper's primary value lies in its relevance to data users, an audience we also wanted to prioritize. Similar papers published in ACP, like Grosvenor & Wood (2014), which examines how viewing geometry biases cloud optical depths, further affirmed our journal choice.**

**LiDAR and AOD.**

I come from the passive remote sensing side, so this may just be me, but I could have used a little bit more depth on AOD products from CALIOP, or maybe a little bit more information up front.

L30 "also provide vertically resolved extinction profiles".   CALIOP measures backscattering profiles, not extinction.

**changed lines 29-30 from "lidar instruments also provide" to** *"lidar products also provide vertically resolved extinction coefficient profiles ($\sigma_{ext}$), representing local light attenuation along the vertical column."*

L44  "lidar ratio"  it is mentioned here for the first time with no explanation.  Perhaps it should be defined?

**updated (lines44-47):**

*"This lidar ratio (S) relates the observed backscatter (β) to an extinction coefficient ($\sigma_{ext}$):*

*$S = \sigma_{ext}/\beta$,*

*where S is given in sr, $\sigma_{ext}$ in km⁻¹, and β in km⁻¹ sr⁻¹. Accordingly, accurate representation of AODs depends on selecting the correct lidar ratio, and so potential errors in aerosol subtyping are a source of uncertainty."*

L111 and L112 "CALIOP retrieves backscatter with depolarization at 532 and 1064 nm, with L2 and L3 aerosol data products providing AODs at 532 nm." Yes. CALIOP retrieves backscatter and depolarization, but then there is a big leap to AOD.

**updated (lines 114-115) to:**

*"CALIOP retrieves backscatter with depolarization at 532 and 1064 nm. Subsequent processing derives aerosol subtype classifications, $\sigma_{ext}$ profiles, and AOD at 532 nm, which are available in both the L2 and L3 aerosol products."*

Eventually the manuscript does describe the CALIOP processing routines, briefly, and it does mention the possibility of incorrect assumptions of lidar ratio affecting biases and correlation between CALIOP AOD and passive sensors. The authors aren't amiss here. I just encourage them to consider bringing a little bit more explanation up front.

**Thank you for the feedback above. We've adapted the manuscript as outlined above, and especially provided more discussion of lidar ratios and associated potential biases in the introduction.**

**MODIS quality flags**

It is no surprise that the sensors develop biases and lose correlation as the number of retrievals move to marginal QA flags (QA = 1). There is confusion in the recommendations for use of data with these QA flags.

On this web page:
https://darktarget.gsfc.nasa.gov/products/viirs-modis/level-2-product-contents

We see the statement:

"For Ocean based products we suggest using only QA 2 and 3"

But on this web page:
https://darktarget.gsfc.nasa.gov/what-are-quality-flags-qa-what-do-they-mean-and-where-can-i-find-them

We see the statement:

"For ocean products we advise using anything above QA zero"

This inconsistency on recommendation is troubling. I, myself, have fallen into the "anything above QA=0" camp and made that recommendation many times. However, even so, the QA=1 designation was put there for a reason.

It would be helpful for this paper to describe the quality flags and describe exactly what are the criteria that would create a QA=1 and then ask why so many low QA in Arctic oceans. One of the criterion might be solar zenith angle itself or a proxy for it. Therefore, QA flag and solar zenith angle are not independent factors and the analysis presented in the paper should clearly explain the overlap and consequences of the overlap. Figures like Figure 7 might be pre-ordained if these parameters are not independent, for example. I noticed that the other reviewer had similar questions about the QA flags.

**Both reviewers requested more information on the QAC flag designation process, and we especially appreciate this feedback as we think these additions will strengthen the manuscript . To address this issue, we added a new subsection (2.4.2) on Quality Flags to the MODIS data section, which we included in our response to Reviewer 1. This section explains in greater detail how various run time flags that arise before or during retrieval processing are used to assign QAC scores. We also added to the supplement a table summarizing the specific criteria that can lead to different QAC flags.**

**As for why low-quality retrievals are so common in the Arctic ocean during winter—we are not entirely sure. While QAC flags do not depend explicitly on the SZA, Fig. 9 shows that higher SZAs clearly correlate with a higher fraction of low-quality retrievals. However, many high-quality retrievals still occur even at very high SZAs (from 75°-80° SZA just under 15% of retrievals are flagged as "high-quality"), which indicates that the SZA alone is not sufficient to result in a "low-quality" designation on its own. Reviewer 1's recommendation that we look more into the bitmasks/run time flags associated with the low-quality retrievals would help us learn more; in lines 588-593 of the conclusion we describe the analysis that should be done:**

Next, the specific conditions responsible for this relationship have yet to be fully described. As described in 2.4.2, low-quality scores over ocean may be assigned due to a number of different retrieval characteristics, and we don't yet know whether the SZA-dependence is evident in all low-quality retrievals. One well-known source of uncertainty in remote sensing is cloud adjacency, whereby shadows or scattered light from neighboring clouds introduce unconstrained effects on reflectance measures in nearby pixels (see Fig. 10). Assessing how different pathways to low QAC scores vary in prevalence and SZA dependence will help clarify the role of this and other 3D artifacts.

Figures like Figure 7 might be pre-ordained if these parameters are not independent

**We agree this may be the case for Fig. 7a. However, Fig. 7c demonstrates that the dependency on the SZA is evident even when controlling for the increased frequency of low-quality retrievals at higher SZAs. This points us to a more complex relationship with the SZA that we originally imagined: low-quality retrievals are both more frequent at high-SZAs, *and* the magnitude of low-quality retrievals declines systematically with higher SZAs. So the unrealistic declines in winter and shoulder-season Arctic AODs in Fig. 5 are due to both an increase in the number of low-quality retrievals, and an increasingly low-bias in low-quality retrievals when SZAs are high.**

**A better understanding of what mechanism, precisely, is driving the bias is an interesting topic that should be explored further. Reviewer 1 mentioned examining the run time flags/bitmasks. As described above, we agree that future work should focus on understanding:**

**1) how the frequencies of all run time flags leading to QAC = 1 varies with the SZA, and also in space and time, and**

**2) to what extent the dependency shown in Fig. 7c is evident for all run time flags leading to QAC=1**

**It's the quality, not the geometry.**

The final conclusion is stated on L512-L513

*However, where sufficient coverage with high-quality*

*L2 MODIS AODs is available, such retrievals may provide useful information even under very high (>70$_o$) SZAs.*

The way I read this paper is that solar zenith angle is NOT the reason for the decoupling of passive and active sensor AOD seasonal cycles. It is the QA flags of the passive sensors. Am I wrong?

**We argue that the decoupling is not from either retrieval quality or the SZA alone, but rather the two together—AODs from low-quality (and only low-quality) retrievals decline as the SZA increases. Fig. 7c shows how this relationship persists without regards to changes in the relative frequency of high- or low-quality retrievals with the SZA.**

**Based on this analysis, we would expect to see decoupling of seasonality even in the absence of seasonal variations in data quality, where data quality is consistently low (such as the Southern Ocean—though more work examining whether this**

**relationship persists across all pathways to low-quality flags would be necessary to say this definitively).**

**Our updates to the manuscript providing more detail about how QAC flags are assigned, and affirming that they do not explicitly depend on the SZA, should help to clarify this point in the manuscript. In addition, we refined several paragraphs in the conclusion to make this point more explicit, which are evident in the updated manuscript.**

This reinforces the need for a more complete description of what triggers a QA flag to go from "good" to "marginal".

**agreed—see above**

Also the authors might want to think about the title again.  It's not really about solar zenith angle in the end, although it made sense to explore the possibility initially.

**We updated the title to emphasize that it is the interplay of retrieval quality and SZA effects that contribute to the bias between datasets:**

*"Seasonality Biases Arise from the Interplay of Retrieval Quality and Solar Zenith Angle Effects in Passive Sensor AODs"*

**Validation against ground truth**

There is none. CALIOP is being used as ground truth but is not.  CALIOP must make a leap from measurements of backscattering profiles to integrated extinction using assumptions of lidar ratios based on aerosol typing.

**We updated the manuscript to clarify that neither CALIOP or MODIS offer ground truth measures, and to better explain that our rationale for normalizing to the mean of the two instruments reflects our assumption that either could contribute to the observed bias. While CALIOP does not provide a ground truth measure, it controls for the retrieval geometry, allowing us to examine whether changes in the SZA contribute to the observed bias (lines 385-386):**

*"As neither instrument serves as a ground-truth reference, we normalize to the mean of the two instruments rather than to either individually."*

**we also clarify this again in the conclusion (lines 600-602):**
*Further comparisons of satellite AODs to AERONET, MAN, and ground-based lidar across a range of SZAs can help disentangle the effects of 3D artifacts on these different assumptions, and should be a focus of future investigations.*

L141 states that "Globally, CALIOP AODs have been validated against AERONET…" I was curious about the subset of validation at Arctic sites for both CALIOP and the passive sensors. Is there a quick way of looking at that from literature? I don't expect such a validation in this paper.

**Below, we are sharing some figures adapted from papers that have compared AERONET AODs to CALIOP or MODIS. However, these figures come with some caveats—first, it is unclear during which months of the year AERONET retrievals at Arctic sites were available. If observations mainly occurred during the summer, that would bypass the seasonality issue. Second, in the case of the MODIS comparison, the validation was only done against high-quality retrievals. We adapted the manuscript to elaborate on these points:** "Biases at specific Arctic AERONET sites all fell within this range. However, because AERONET data are unavailable during low-light conditions, the validation may not reflect seasonality-related biases."

**Adapted from Kim et al., (2018). The bias between AERONET and CALIOP at Arctic sites falls between -.15 to -.025, with half the stations falling within the -.075 to -.05 bin. This is broadly consistent with the global error of −0.051 ± 0.08.**

[Figure]

**Figure 16.** Global maps of mean AOD difference between CALIOP and MODIS. CALIOP data for **(a)** V3 and **(b)** V4 and MODIS collection 6 from 2007 to 2009. Mean AOD difference between CALIOP and AERONET is shown in circles.

Adapted from Wei et al., (2019), below: the RMSE and MDB between AERONET and MODIS in the Arctic similarly fell within the global distribution for collection 6.1. MODIS validation against AERONET examines different geographic bins. However Arctic sites are spread between different continental categories, which could obscure differences within the Arctic as a group; future studies could classify the Arctic separately, however we do not see substantial differences in the bias at Arctic versus European/N.American sites. Again, these comparisons were between AERONET and high-quality MODIS retrievals.

[Figure]

Fig. 1. Geographical boundaries of regions defined in this study. Red dots show the locations of AERONET sites. (For interpretation of the references to colour in this figure legend, the reader is referred to the Web version of this article.)

Fig. 4. Validation of Terra MODIS C6.1 DT, DB, and DTB AOD retrievals against AERONET AODs for each site from 2013 to 2017: (a–c) correlation (R), (d–f) percentage of retrievals within the expected error envelopes (%), (g–i) median bias, and (j–m) root-mean-square error.

**In Sayer et al., (2014), validation for collection 6, comparisons between high-quality MODIS AODs and AERONET examine the error against the SZA, finding no dependency at high SZAs. This is consistent with our findings that there is no dependency in the high quality subset.**

[Figure]

**Figure 17.** Median bias in MODIS AOD as a function of (a) surface altitude, (b) solar zenith angle, (c) view zenith angle, and (d) scattering angle, from the "collocated" set of DB/DT/AERONET matchups. DB data are shown with blue diamonds, DT with green triangles, and the merge with red squares. The line of zero difference is dashed.

**Some minor quibbles**

L296-L297. POLDER is also multi angle

L301. What is meant by brightest months of the year.

Authors list. Does co-author Levy really want to be listed as Rob Levy and not Robert C. Levy. It makes future searches more difficult.

**all have been addressed in the revised manuscript**

**Other Changes**

In response to feedback from preliminary reviews, we also changed the spelling of collocations to colocations, and we updated the data availability figure to use a perceptually uniform colormap.